# Independent origin of large labyrinth size in turtles

Serjoscha W. Evers [1,2] ✉, Walter G. Joyce[1], Jonah N. Choiniere[3], Gabriel S. Ferreira [4,5], Christian Foth [1], Guilherme Hermanson [1,6], Hongyu Yi[7,8], Catherine M. Johnson [2], Ingmar Werneburg[4,5] & Roger B. J. Benson [2,3]

The labyrinth of the vertebrate inner ear is a sensory system that governs the perception of head rotations. Central hypotheses predict that labyrinth shape and size are related to ecological adaptations, but this is under debate and has rarely been tested outside of mammals. We analyze the evolution of labyrinth morphology and its ecological drivers in living and fossil turtles, an understudied group that underwent multiple locomotory transitions during 230 million years of evolution. We show that turtles have unexpectedly large labyrinths that evolved during the origin of aquatic habits. Turtle labyrinths are relatively larger than those of mammals, and comparable to many birds, undermining the hypothesis that labyrinth size correlates directly with agility across vertebrates. We also find that labyrinth shape variation does not correlate with ecology in turtles, undermining the widespread expectation that reptilian labyrinth shapes convey behavioral signal, and demonstrating the importance of understudied groups, like turtles.

The vertebrate labyrinth is the sensory 'organ of balance'. Three semicircular canals (SCC) of the cranial skeleton house the membranous semicircular ducts, which detect rotational head accelerations. These provide sensory input on head motion, which, among other functions, enable the vestibulo-ocular (VOR) and vestibulo-collic (VCR) reflexes that help stabilize the visual field during locomotion[1–7]. The lengths and pathways of the semicircular ducts have predictable effects on the sensory response profile, which is often described as 'sensitivity', but can be decomposed into distinct effects such as response time to rotational accelerations, and signal discrimination[3–9]. This has given rise to the hypothesis that labyrinth form varies among species according to their ecological specializations[3,6]. Hypothesized correlations between labyrinth form and locomotor behaviors are particularly widespread, because different locomotor modes imply different head motions, therefore plausibly requiring differences in performance of the vestibular organ (e.g., ref. 4). If these generalizations are indeed true, they should have strong predictive power across vertebrates.

Most empirical tests of the form-function relationships of the SCC have used mammals as a model system[4,10–23]. SCC lengths show strong correlations with head size (or with body mass) in mammals[4,24]. The residuals from this relationship represent variation in the proportional labyrinth size, indexing the relative sensitivity of the labyrinth to signal discrimination, and are correlated with locomotor behavior[11]. Proportionally larger labyrinths in agile mammals[4,11] may be functionally related to higher visual acuities required for more precise gaze

[1]Department of Geosciences, University of Fribourg, Chemin du Musée 6, 1700 Fribourg, Switzerland. [2]Department of Earth Sciences, University of Oxford, South Parks Road, Oxford OX1 3AN, United Kingdom. [3]Evolutionary Studies Institute, University of the Witwatersrand, 1 Jan Smuts Avenue, Johannesburg 2000, South Africa. [4]Senckenberg Centre for Human Evolution and Paleoenvironment an der Universität Tübingen, Sigwartstraße 10, 72076 Tübingen, Germany. [5]Fachbereich Geowissenschaften, Universität Tübingen, Hölderlinstraße 12, 72074 Tübingen, Germany. [6]Laboratório de Paleontologia de Ribeirão Preto, FFCLRP, Universidade de São Paulo, Ribeirão Preto, Brazil. [7]Key Laboratory of Vertebrate Evolution and Human Origins of Chinese Academy of Sciences, Institute of Vertebrate Paleontology and Paleoanthropology, Chinese Academy of Sciences Beijing, 100049 Beijing, China. [8]CAS Center for Excellence in Life and Paleoenvironment Beijing, 100044 Beijing, China. ✉e-mail: serjoscha.evers@unifr.ch

stabilization[25]. The cross-sectional area of the duct lumen influences response time of the labyrinth[5], but has been studied less frequently due to difficulties imaging the membranous ducts compared to the bony canals. Shape aspects such as canal orientations, aspect ratios, and ellipticity have also been hypothesized to contain ecomorphological signals[6,9,18,26,27]. These hypothesized links between SCC shapes and ecology are of growing importance to paleontological studies that attempt to infer ecological traits of extinct species from their labyrinth shape, with implications for deep evolutionary patterns in vertebrates (e.g., refs. [28–31]).

Nevertheless, quantitative studies of non-mammalian vertebrates have challenged even the most widely-accepted hypotheses of ecological signal in labyrinth morphology. For example, statistical analyses detect only weak relationships (if any) between locomotory capabilities and labyrinth shape variation in birds or extinct reptiles after accounting for shared evolutionary history[32–34] (but see ref. [35]), and some other studies regarded potential locomotion-related variation in labyrinth shape as 'secondary variation' compared to the strength of phylogenetic signal (e.g., refs. [6,22]). Furthermore, the proportionally large labyrinths of birds have been hypothesized as a flight-related adaption[24]. However, large labyrinth sizes evolved much earlier, among the non-flying ancestors of birds[34]; and other flying vertebrates (i.e., bats and pterosaurs) have relatively small labyrinths[21,34]. These findings highlight the need for broader scrutiny of the relationship between labyrinth morphology and ecology in vertebrates, and the potential for insight based on analyses of understudied groups.

Turtles provide a powerful test of the predictions of labyrinth ecomorphology because their living and extinct representatives exhibit diverse ecological adaptations and locomotor behaviors, ranging from terrestrial walking to oceanic diving[36,37] throughout 230 million years of evolutionary history[38]. Very little is known about the evolution of the turtle labyrinth. Some studies suggest that turtles have small labyrinth sizes[39], possibly consistent with their low agility and slow movement speeds. However, this is based on comparative measurements of a single species of giant terrestrial tortoise (*Aldabrachelys gigantea*)[24] as well as qualitative comparisons between turtles and other amniotes[39].

Here, we show that turtles have large relative labyrinth sizes that evolved independently to large labyrinths in other major vertebrate groups. Using phylogenetic comparative methods, we show that labyrinth size is correlated with ecology in turtles, whereby small labyrinth sizes are associated with terrestrial habits. Large relative labyrinth sizes evolved early on the turtle stem lineage, during the aquatic diversification of the group, and possibly triggering important skull modifications related to the adductor muscle redirecting system of turtles. We also show that labyrinth shape variation, quantified using 3D characterizations of SCC geometry, cannot be explained by ecology or neck function, and is instead best explained by allometry and spatial constraints of braincase morphology.

## Results

### Labyrinth shape variation in turtles

Principal component analysis (PCA) ordination of labyrinth landmark data illustrates major aspects of shape variation among living and extinct turtle labyrinths. The first three principal components (PCs) collectively summarize 54.5% of labyrinth shape variance, and only the first five axes each explain >5% of the variance. Ecological groups have considerable overlap in the morphospace defined by PC axes 1–3 (Fig. 1; Supplementary Fig. 4). PC1 summarizes 29.4% of total labyrinth shape variation, with negative values indicating dorsoventrally high and anteroposteriorly short labyrinths with thick SSCs, as seen in the leatherback sea turtle *Dermochelys coriacea* (Fig. 1a, c; Supplementary Figs. 5–8). Positive values of PC1 indicate dorsoventrally low and anteroposteriorly long labyrinth morphology with slender SSCs as seen in chelids and trionychians (Fig. 1a, c; Supplementary Figs. 5–8).

PC2 explains 15.9% of shape variation, with negative values describing dorsoventrally tall, transversely wide labyrinths (e.g., *Australochelys africanus*) and positive values corresponding to anteroposteriorly elongate and mediolaterally narrow shapes seen in many testudinoids (Fig. 1a, c; Supplementary Figs. 5–8) and the early stem turtle *Proganochelys quenstedtii* (Fig. 1a; Supplementary Figs. 5–8). PC3 accounts for 9.3% of labyrinth shape variation, with negative values describing a relatively wide LSC combined with short, dorsoventrally low, and thin SSCs (e.g., the chelydrid *Macrochelys temminckii* Fig. 1b, d; Supplementary Figs. 5–8), and positive values describe a dorsoventrally high but mediolaterally narrow morphology with increased canal diameters (e.g., in *D. coriacea* and testudinids; Fig. 1b, d; Supplementary Figs. 5–8).

Our PCA plots show some separation of labyrinth shapes among turtles with different ecologies, in spite of overlaps (Fig. 1). However, PCA itself does not provide a statistical test of the potential explanations for that separation. Apparent clustering of labyrinth shapes by ecology in Fig. 1 could therefore be explained either by direct effects (i.e., that labyrinth shape has a direct functional relationship with ecology) or by indirect effects, such as phylogenetic autocorrelation (e.g., ref. [40]), including a 'spatial constraints' hypothesis, in which braincase shape and head/body size vary among groups with different ecologies, and have an important influence on labyrinth shape (e.g., ref. [34]). We show that those indirect effects are stronger than any direct ecological signal using shape regressions, below. We also illustrate that here by showing substantially less ecological separation when PCA ordination is applied to shape data after correction for the effects of skull size and braincase aspect ratio (using residuals from shape regressions, below) (Fig. 2; Supplementary Fig. 9).

### Labyrinth shape regressions

Head size, labyrinth size and braincase aspect ratio have significant effects on labyrinth shape in Procrustes distance-based phylogenetic regressions (Table 1). This indicates important effects of allometry and spatial constraint on labyrinth shape (Fig. 2e, f). In contrast, we find no significant influence of ecology on labyrinth shape in any model, even in simple models of the form: shape ~ ecology. Therefore, the graphical association between locomotion and labyrinth shape in our PCA morphospace (Fig. 1) is not explained by direct functional linkage but by other factors (e.g., variation in soft-tissue structure of the vestibular organ, allometry, braincase architecture, or other uncharacterized aspects that are embodied by 'phylogenetic signal' in our analysis). We also find no relationship of labyrinth shape to other traits related to habitual head motion (e.g., neck retraction ability and type). The best model in which all variables are significant combines independent effects of allometric and spatial constraint variables with an interaction term between skull size and braincase aspect ratio (i.e., labyrinth shape ~ skull size × braincase aspect ratio + labyrinth size; Table 1), in total accounting for 14.7% of labyrinth shape variance (Table 1). This indicates substantial unexplained variation in labyrinth shape. The interaction term in this model indicates that the effect of braincase aspect ratio on labyrinth shape varies depending on skull size. In this and other models, effect size ($Z$ scores) of the spatial constraint variable exceed those of allometric variables (Table 1). Labyrinth shape deformation associated with skull size show that large-headed turtles have labyrinths with decreased ASC and LSC lengths (Fig. 2e). High braincase aspect ratios are associated with high aspect ratios of the labyrinth, in which the vertical SCC are dorsoventrally tall, whereas the labyrinth is mediolaterally narrow and anteroposteriorly short (Fig. 2f). Our conclusions are robust to sensitivity analyses, including: deletion of marine turtles to test the influence of derived vestibular morphology in chelonioids (Supplementary Tables 4–5); and deletion of landmarks around

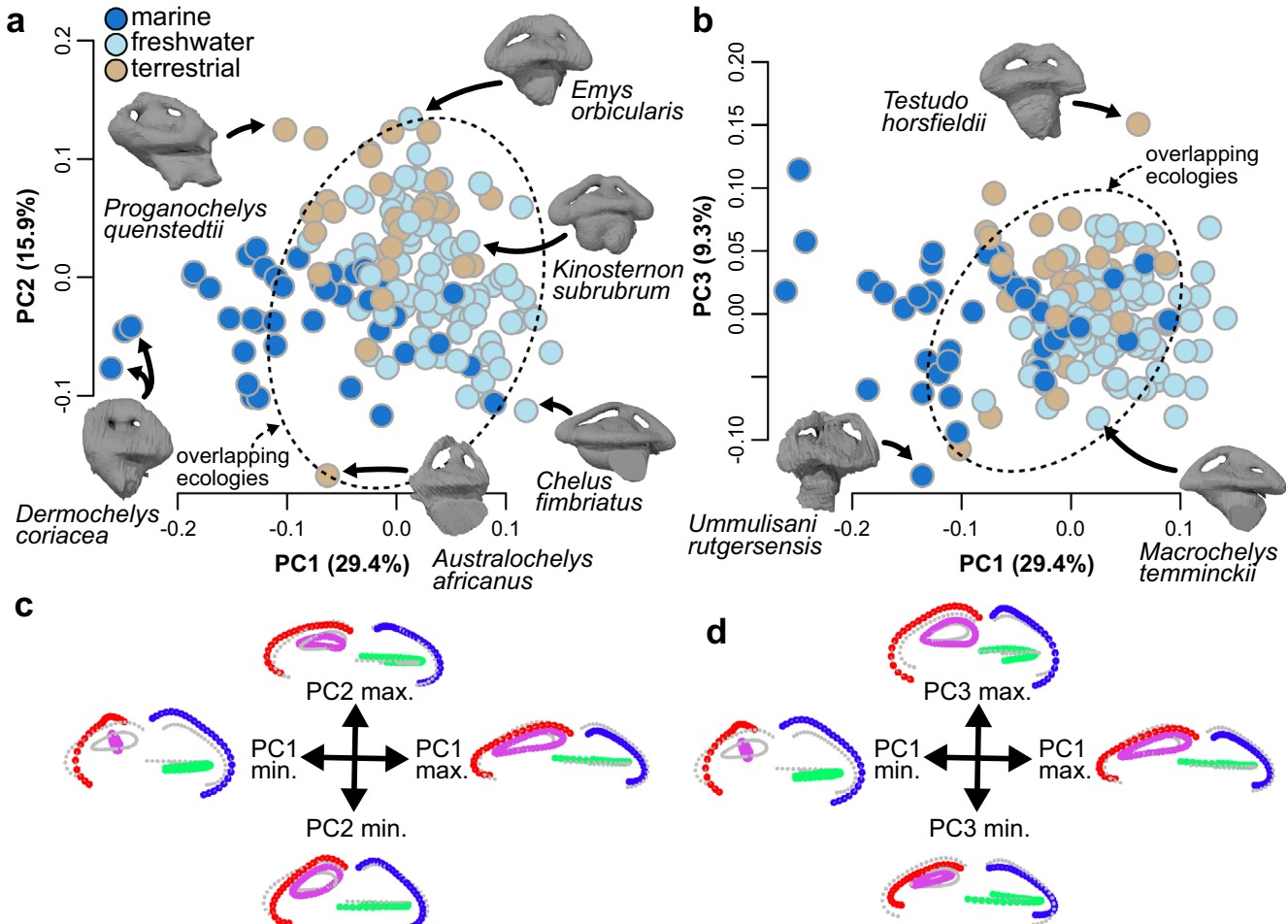

**Fig. 1 | Morphospace of turtle labyrinth shape from PCA ordination ($N = 168$).**
**a** PC1 vs. PC2. **b** PC1 vs. PC3. Dashed ovals in **a** and **b** show overlapping ecologies. **c** Maximum shape deformations of the landmark data in lateral view along PC1–2 against mean shape. **d** Maximum shape deformations of the landmark data in lateral view along PC1–3 against mean shape. Proportions of total shape variance explained by PC axes are given in brackets. For a version with taxon point labels, see Supplementary Fig. 4. Source data are provided with this paper.

the inner perimeter of the anterior SCC to test the influence of SCC thickness variation (Supplementary Table 6).

**Labyrinth size regressions**
Contrary to labyrinth shape, labyrinth size variation in turtles can be explained by ecological effects, at least in part. Phylogenetic generalized least squares regression (pGLS) model comparisons demonstrate independent effects of skull size and habitat ecology (terrestrial | aquatic), whereby terrestrial species have proportionally small labyrinths in relation to head size (Table 2). The best model according to Akaike's Information Criterion for finite sample sizes (AICc)[41] takes the form: labyrinth size ~ Skull box volume + forelimbs not-webbed (AICc weight = 0.152; $R^2 = 0.897$; Table 2). The coefficient of 'forelimbs not-webbed' in this model (slope$_{notwebbed}$ = −0.07, SE$_{coef}$ = 0.02, $p < 0.01$; Table 2) indicates that terrestrial turtles have smaller labyrinth sizes, where the degree of forelimb webbing is positively correlated with increasing aquatic adaptation in turtles[42] (see methods). However, this effect is weak, explaining only 2.8% of residual variation after accounting for skull size (Table 2).

Among models with non-negligible AICc weights, variables associated with high degrees of aquatic adaptation ('open water', 'freshwater', 'extremely webbed') are significantly associated with increases in labyrinth size, and variables indicating high levels of terrestrial adaptation ('forelimbs not-webbed', 'terrestrial') are significantly associated with decreasing labyrinth size (Table 2).

Terrestrial and aquatic effects are independent of one another, as they retain individual significance when included in the same model (Table 2). These results specifically suggest that actively swimming freshwater aquatic turtles have large labyrinths, contrary to previous findings that some aquatic amniotes have small labyrinths (cetaceans[10]; eosauropterygians[43]; Ross seals[44]). A significant effect of the braincase aspect ratio is also returned in many models with non-negligible AICc weights. Labyrinth size decreases with increasing braincase aspect ratios (i.e., in species with less flat skulls), although this effect is small, explaining 1.8% of residual size variation after accounting for skull size. The effects of neck retraction ability and the type of neck movements are insignificant and receive negligible AICc support in bivariate and multiple regressions. The effect of relative neck lengths could only be tested on a smaller dataset ($N = 56$), but neck length variables are neither significant on their own, nor in multiple regression models, and all models with neck length variables included receive negligible AICc support (Supplementary Data 14–15).

Skull size is important in all explored models ($p < 0.001$), regardless of which skull size index is employed, and exhibits high correlation coefficients with labyrinth size (labyrinth size ~ head size: $R^2 = 0.821$ to 0.894; Supplementary Data 14–15). The relationship between labyrinth size and skull size includes strong levels of phylogenetic signal ($\lambda = 0.890$ in the best model) and has a slope that indicates strong

negative allometry (slope = 0.23 in best model; compared to an iso-metric slope of 0.33). The residuals from this relationship describe variation in relative labyrinth size from expectations based on head size and were mapped to phylogeny in Fig. 3. Most major turtle lineages have large residual variation and include species with both positive and negative residuals (Fig. 3). However, large relative labyrinth sizes are

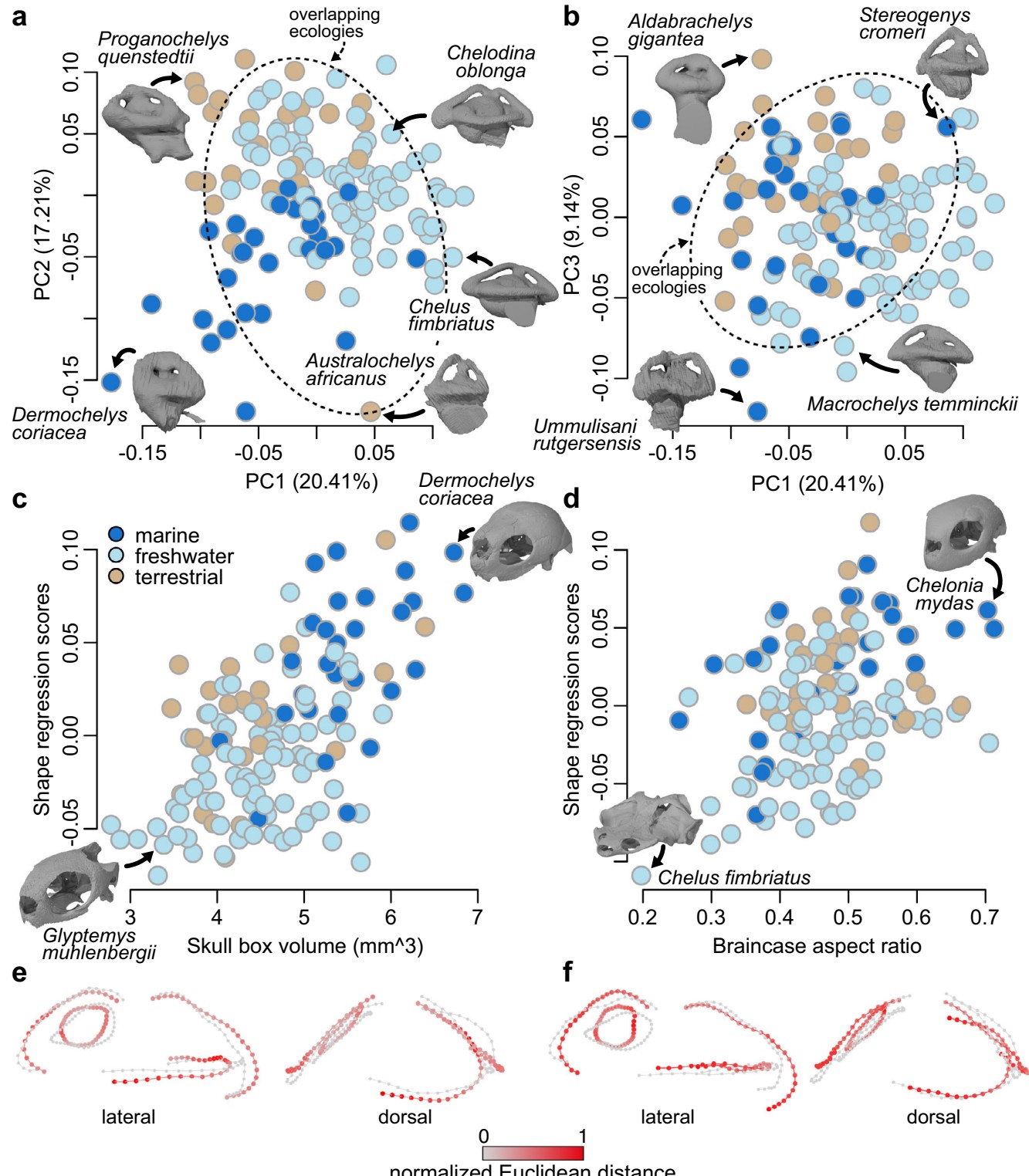

**Fig. 2 | Morphospaces of labyrinth shape corrected for size and aspect ratio (N = 138) and allometric and braincase aspect ratio effects on labyrinth shape.** **a** PC1 vs. PC2 of labyrinth shape corrected for skull size allometry and braincase aspect ratio (**b**) PC1 vs. PC3 of labyrinth shape corrected for skull size allometry and braincase aspect ratio. Proportions of total shape variance explained by PC axes is given in brackets. Dashed ovals in **a** and **b** show overlapping ecologies. **c** Plots of skull box volume regression scores against skull box volume, using the formula 'shape - skull box volume + braincase aspect ratio'. **d** Plots of braincase aspect ratio regression scores against skull box volume. **e** Deformation plots at large (red) and small (gray) skull sizes. **f** Deformation plots at large and small brain aspect ratios. For a version with taxon point labels, see Supplementary Fig. 9. Source data are provided with this paper.

**Table 1 | Results of selected phylogenetic Procrustes distance regressions of labyrinth shape ~ independent variables including fossils**

| Model | Variable | Effect | F statistic | Z score | P value | $R^2$ | $R^2$ model | re-df |
|---|---|---|---|---|---|---|---|---|
| ~Skull box volume *braincase aspect ratio + labyrinth centroid size | Skull box volume | Allometric | 6.365 | 4.143 | 0.001 | 0.040 | 0.147 | 133 |
| | Braincase aspect ratio | Spatial constraint | 7.118 | 4.457 | 0.001 | 0.045 | | |
| | Labyrinth centroid size | Allometric | 4.403 | 3.351 | 0.002 | 0.028 | | |
| | Skull box volume: labyrinth centroid size | Interaction | 5.396 | 3.572 | 0.001 | 0.034 | | |
| ~Skull box volume *braincase aspect ratio | Skull box volume | Allometric | 5.049 | 3.690 | 0.001 | 0.032 | 0.122 | 134 |
| | Braincase aspect ratio | Spatial constraint | 7.611 | 4.500 | 0.001 | 0.049 | | |
| | Skull box volume: braincase aspect ratio | Interaction | 6.317 | 3.746 | 0.001 | 0.041 | | |
| ~Skull box volume *labyrinth centroid size | Skull box volume | Allometric | 7.803 | 4.340 | 0.001 | 0.052 | 0.112 | 134 |
| | Labyrinth centroid size | Allometric | 5.814 | 3.888 | 0.001 | 0.039 | | |
| | Skull box volume: labyrinth centroid size | Interaction | 3.089 | 2.349 | 0.012 | 0.021 | | |
| ~Braincase aspect ratio | Braincase aspect ratio | Spatial constraint | 9.792 | 4.750 | 0.001 | 0.067 | 0.067 | 136 |
| ~Skull height | Skull height | Allometric | 9.533 | 4.702 | 0.001 | 0.066 | 0.066 | 136 |
| ~Skull box volume | Skull box volume | Allometric | 7.266 | 4.255 | 0.001 | 0.051 | 0.051 | 136 |
| ~Skull length | Skull length | Allometric | 6.513 | 4.073 | 0.001 | 0.046 | 0.046 | 136 |
| ~Skull width | Skull width | Allometric | 5.549 | 3.760 | 0.001 | 0.039 | 0.039 | 136 |
| ~Labyrinth centroid size | Labyrinth centroid size | Allometric | 5.299 | 3.604 | 0.001 | 0.037 | 0.037 | 136 |
| ~Skull box volume + marine.all | Skull box volume | Allometric | 6.792 | 4.022 | 0.001 | 0.047 | 0.058 | 135 |
| | All extant and extinct marine taxa | Ecological | 1.543 | 1.140 | 0.125 | 0.011 | | |
| ~Marine.all | All extant and extinct marine taxa | Ecological | 1.963 | 1.578 | 0.056 | 0.014 | 0.003 | 136 |
| ~Marine.extant | Extant marine taxa | Ecological | 0.993 | 0.287 | 0.399 | 0.007 | 0.005 | 136 |
| ~Freshwater | Freshwater habitat ecology | Ecological | 1.500 | 1.057 | 0.150 | 0.001 | 0.012 | 136 |
| ~Terrestrial | Terrestrial habitat ecology | Ecological | 1.351 | 0.814 | 0.202 | 0.010 | 0.014 | 136 |
| ~Incomplete retraction | No ability to retract neck | Morpho-functional | 1.420 | 0.954 | 0.182 | 0.010 | 0.010 | 136 |
| ~Full retraction | Ability to fully retract neck | Morpho-functional | 1.420 | 0.954 | 0.182 | 0.010 | 0.010 | 136 |
| ~No retraction plane | No preferred plane developed, ancestral anatomy | Morpho-functional | 0.341 | −1.611 | 0.945 | 0.003 | 0.003 | 136 |
| ~Vertical retraction | Cryptodiran neck anatomy | Morpho-functional | 0.235 | −2.535 | 0.999 | 0.002 | 0.002 | 136 |
| ~Horizontal retraction | Pleurodiran neck anatomy | Morpho-functional | 0.550 | −0.980 | 0.834 | 0.004 | 0.004 | 136 |

$N = 138$ for all models. Best model is on top. Hypothesis testing used a phylogenetic Procrustes ANOVA[89], in which statistical significance (unadjusted $P$ values) is calculated by comparison of phylogenetically-transformed sum-of-squared Procrustes distances with sum-of-squares distributions generated from residual randomization permutation procedure (RRPP)[91], using 1000 permutations. $F$-statistic is the ratio between the sum of squares of the regression and the sum of squares of the error. Effect sizes ($Z$ scores) were computed as standard deviations of $F$-distributions using residual degrees of freedom (re-df).

present in species of several groups, including chelydroids, which include the turtle with the largest relative labyrinth size, *Dermatemys mawii*.

## Labyrinth size evolution of turtles

Testudinidae (tortoises) are the only group to show consistently negative labyrinth size residuals (Fig. 3b), consistent with the finding of small labyrinth size in terrestrial species (above). Testudinids include the species that shows the smallest relative labyrinth size of all turtles measured, *Aldabrachelys gigantea*. This species is the only turtle for which relative labyrinth sizes had been reported previously[24]. Early stem-group turtles including the Triassic *Proganochelys quenstedtii* and the Jurassic *Australochelys africanus* and *Kayentachelys aprix* also have small labyrinths, indicating ancestrally small relative labyrinth sizes, supporting terrestrial behaviors of these early stem turtles (Fig. 3b, c). Large relative labyrinth sizes first evolved in the Middle Jurassic, as exemplified by perichelydian stem turtles including *Eileanchelys waldmani*, *Kallokibotion bajazidi*, and paracryptodires (Fig. 3b, c), and coincides with the diversification

into aquatic habitats among these turtles as well as a size increase of the otic capsule that resulted in the evolution of trochlear mechanisms for redirecting jaw musculature[45,46].

## Amniote labyrinth sizes

Comparative size data for tetrapods shows that most turtles have larger relative labyrinth sizes than those of mammals and many reptiles (Fig. 4; Supplementary Fig. 10). This contrasts with what would be expected under a general 'agility' hypothesis for explaining labyrinth size variation among tetrapods, or even just among reptiles. Although some bird species have relative labyrinth sizes that exceed those of turtles, the distribution of turtle (and lepidosaur) labyrinths overlaps strongly with that of birds when compared to postrostral skull length (Fig. 4). Extant testudinids, on the other hand, have some of the smallest relative labyrinth sizes among reptiles (Fig. 4). Among extant non-turtle reptiles, only crocodylians have proportionally small labyrinths, similar to or only slightly larger than the ancestral small labyrinth sizes of reptiles. The existence of a small ancestral labyrinth size is indicated by the small relative labyrinth sizes of stem archosaurs, stem crocodylians[34], and early stem turtles (Fig. 4).

**Table 2 | Results of pGLS regressions of labyrinth centroid size ~ independent variables for extant turtles, showing only models with non-negligible AICc values (full table in Supplementary Data 14)**

| Independent variables | λ | AICc | AICc weight | $R^2$ | re-df | Variable | Coefficient | t value | P value |
|---|---|---|---|---|---|---|---|---|---|
| ~Skull box volume + forelimbs not webbed | 0.890 | −266.74 | 0.147 | 0.897 | 86 | Intercept | 0.50 | 12.523 | $4.4 \times 10^{-21}$ |
| | | | | | | $Log_{10}$(skull box volume) | 0.23 | 23.691 | $5.7 \times 10^{-47}$ |
| | | | | | | forelimbs not webbed | −0.07 | −3.336 | 0.001 |
| ~Skull box volume + braincase aspect ratio + open water locomotion | 0.972 | −266.56 | 0.135 | 0.899 | 85 | Intercept | 0.61 | 10.179 | $2.2 \times 10^{-16}$ |
| | | | | | | $Log_{10}$(skull box volume) | 0.21 | 25.460 | $1.5 \times 10^{-41}$ |
| | | | | | | braincase aspect ratio | −0.19 | −2.892 | 0.005 |
| | | | | | | open water locomotion | 0.05 | 3.099 | 0.003 |
| ~Skull box volume | 0.993 | −266.39 | 0.124 | 0.894 | 87 | Intercept | 0.50 | 11.705 | $1.5 \times 10^{-19}$ |
| | | | | | | $Log_{10}$(skull box volume) | 0.23 | 29.216 | $7.1 \times 10^{-42}$ |
| ~Skull box volume + braincase aspect ratio | 0.983 | −265.89 | 0.097 | 0.896 | 86 | Intercept | 0.60 | 9.577 | $3.3 \times 10^{-15}$ |
| | | | | | | $Log_{10}$(skull box volume) | 0.22 | 25.501 | $7.1 \times 10^{-43}$ |
| | | | | | | braincase aspect ratio | −0.16 | −2.306 | 0.023 |
| ~Skull box volume + freshwater habitat ecology | 0.876 | −265.75 | 0.090 | 0.895 | 86 | Intercept | 0.44 | 10.688 | $1.9 \times 10^{-17}$ |
| | | | | | | $Log_{10}$(skull box volume) | 0.23 | 30.055 | $2.2 \times 10^{-47}$ |
| | | | | | | Freshwater habitat ecology | 0.05 | 3.380 | 0.001 |
| ~Skull width | 0.985 | −265.70 | 0.088 | 0.893 | 87 | Intercept | 0.52 | 12.419 | $5.7 \times 10^{-21}$ |
| | | | | | | $Log_{10}$(skull width) | 0.63 | 28.606 | $5.0 \times 10^{-46}$ |
| ~Skull box volume + open water locomotion | 0.986 | −264.34 | 0.045 | 0.894 | 86 | Intercept | 0.48 | 11.660 | $2.1 \times 10^{-19}$ |
| | | | | | | $Log_{10}$(skull box volume) | 0.22 | 29.214 | $2.0 \times 10^{-46}$ |
| | | | | | | open water locomotion | 0.05 | 2.540 | 0.013 |
| ~Skull box volume + braincase aspect ratio + forelimbs not webbed | 0.935 | −264.32 | 0.044 | 0.896 | 85 | Intercept | 0.59 | 9.644 | $2.7 \times 10^{-15}$ |
| | | | | | | $Log_{10}$(skull box volume) | 0.22 | 26.479 | $8.0 \times 10^{-43}$ |
| | | | | | | braincase aspect ratio | −0.13 | −1.862 | 0.066 |
| | | | | | | forelimbs not webbed | −0.06 | −2.776 | 0.007 |
| ~Skull box volume + terrestrial habitat ecology | 0.920 | −264.27 | 0.043 | 0.894 | 86 | Intercept | 0.50 | 12.289 | $1.3 \times 10^{-20}$ |
| | | | | | | $Log_{10}$(skull box volume) | 0.23 | 29.467 | $1.0 \times 10^{-46}$ |
| | | | | | | terrestrial habitat ecology | −0.04 | −2.885 | 0.005 |
| ~Skull box volume + braincase aspect ratio + freshwater habitat ecology | 0.937 | −263.50 | 0.029 | 0.895 | 85 | Intercept | 0.55 | 8.577 | $3.9 \times 10^{-13}$ |
| | | | | | | $Log_{10}$(skull box volume) | 0.22 | 26.585 | $5.9 \times 10^{-43}$ |
| | | | | | | braincase aspect ratio | −0.13 | −1.936 | 0.056 |
| | | | | | | Freshwater habitat ecology | 0.04 | 2.776 | 0.007 |
| ~Skull box volume + braincase aspect ratio + open water locomotion + forelimbs not webbed | 0.928 | −263.16 | 0.025 | 0.898 | 84 | Intercept | 0.60 | 10.204 | $2.3 \times 10^{-16}$ |
| | | | | | | $Log_{10}$(skull box volume) | 0.22 | 26.152 | $4.0 \times 10^{-42}$ |
| | | | | | | braincase aspect ratio | −0.16 | −2.434 | 0.017 |
| | | | | | | open water locomotion | 0.05 | 2.770 | 0.007 |
| | | | | | | Forelimbs not webbed | −0.05 | −2.337 | 0.022 |
| ~Skull box volume + braincase aspect ratio + terrestrial habitat ecology | 0.945 | −263.09 | 0.024 | 0.895 | 85 | Intercept | 0.60 | 9.840 | $1.1 \times 10^{-15}$ |
| | | | | | | $Log_{10}$(skull box volume) | 0.22 | 26.280 | $1.5 \times 10^{-46}$ |
| | | | | | | braincase aspect ratio | −0.15 | −2.166 | 0.033 |
| | | | | | | terrestrial habitat ecology | −0.04 | −2.637 | 0.010 |
| ~Skull box volume + forelimbs extremely webbed | 0.992 | −262.12 | 0.016 | 0.891 | 86 | Intercept | 0.49 | 11.731 | $1.5 \times 10^{-19}$ |
| | | | | | | $Log_{10}$(skull box volume) | 0.23 | 29.326 | $1.5 \times 10^{-46}$ |
| | | | | | | forelimbs extremely webbed | 0.05 | 1.864 | 0.066 |

Models are ordered by AICc rank, showing the best model on top. N = 89 for all models. λ (lambda) is the phylogenetic signal[102] and was estimated during model fitting. $R^2$ is the generalized coefficient of determination described by ref. 100. Coefficients are estimated using pGLS restricted maximum likelihood. The t statistics are coefficient estimates divided by their standard error. P values are two-sided and unadjusted, and are calculated using the coefficient value and a t-distribution with the number of residual degrees of freedom (re-df) of the model.

## Discussion

Turtles have a conservative labyrinth shape that is already apparent early on in their stem lineage, characterized by a dorsoventrally compact morphology with near-symmetrical SCC. This contrasts with the stem lineages of other reptiles, including those of archosaurs, crocodiles, and birds, in which some members show high morphological diversity and distinct morphologies relative to their crown groups[34].

We find no support for the locomotor hypothesis of variation in labyrinth shape among turtles. Locomotor ecology has been assumed to have a notable influence on labyrinth shape by many previous studies[9,18,20,22,27,35], although alternative hypotheses have rarely been

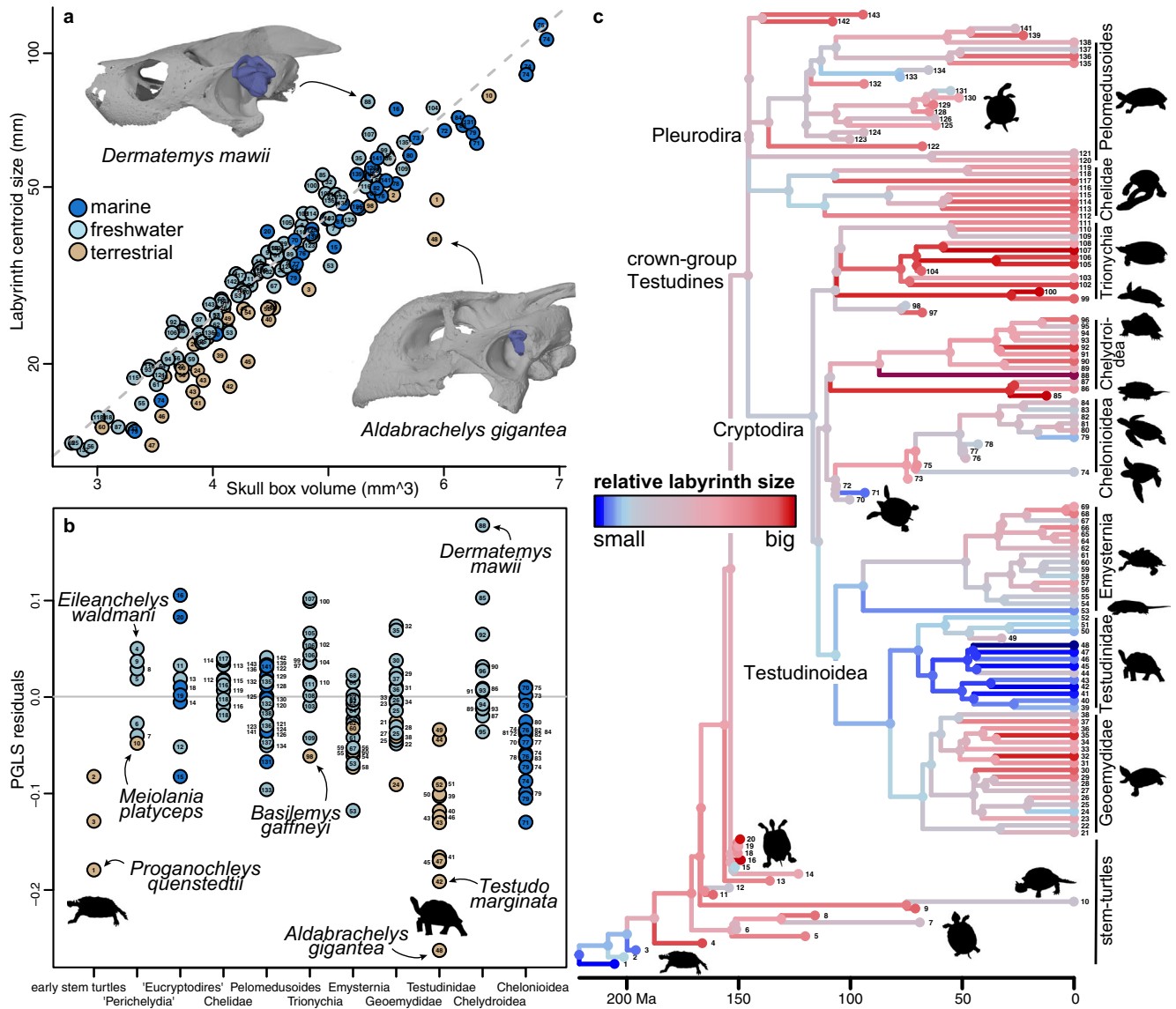

**Fig. 3 | Relative labyrinth size in turtles. a** PGLS regression of labyrinth size on skull box volume. Note that duplicate specimens for several species have been projected into graph but are not part of formal regression. Images show labyrinth models superimposed in original scale to skull models, with skull sizes of both specimens scaled to same length. **b** Residual plot of **a**, with specimens ordered according to major clades. **c** evolution of relative labyrinth size (scaled between 0

and 1 residual labyrinth size) across time-scaled turtle phylogeny, with ancestral values estimated using the ace function of ape 5.3[76]. "Ma" is "million years before present". Numbers in **a**, **b** match tip labels of **c**. For a legend of tip labels, see Supplementary Data 1 and Supplementary Fig. 4. Source data are provided with this paper.

tested. Furthermore, many of the interpretations of previous studies have been based on non-phylogenetic analyses (e.g., refs. 22,35), often overlooking the low explanatory power or absence of relationships between locomotion and labyrinth shape once phylogeny is taken into account (see phylogenetic regressions in refs. 22,35 and recently explained in ref. 47).

We find that variation in the aspect ratio of the braincase is the primary influence on labyrinth shape in turtles, with low braincases being associated with low labyrinth morphologies (Fig. 2f). This supports some recent studies that suggested that labyrinth shape may largely be determined by braincase architecture[27,32,34] (but see ref. 35), with a few exceptions for specific shape aspects[34,43,48]. Hansen et al.[35] recently dismissed the effects of allometry and spatial constraints on labyrinth shape for reptiles and birds. However, allometric effects on labyrinth shape (body mass vs. labyrinth shape $R^2 = 0.22$)[35] explain twice as much variance as the most important ecological variables in their phylogenetic regressions ($R^2 = 0.11$; Table S4 of ref. 35), and their

analyses did not investigate whether independent effects of ecology remained after accounting for braincase architecture. Hansen et al.[35] instead suggested that ecological signal exerts strong functional selection on labyrinth shape, and that resulting variation in labyrinth shape causes variation in adult braincase morphology due to developmental linkage[35,49]. This hypothesis is causally the opposite of a 'spatial constraints' hypothesis, in which variation in braincase morphology is under stronger functional selection and causes variation in labyrinth shape[34] (also due to developmental linkage[49]). We regard the spatial constraints hypothesis as being more plausible for several reasons: (1) published analyses so far, including ours, report at most a weak correlation of labyrinth shape to ecological traits[18,21,22,34], suggesting little functional selection for labyrinth shape; (2) in the few cases examined so far, the effects of braincase shape remain after accounting for ecological effects, but the effects of ecological traits do not remain after accounting for braincase shape[34], suggesting primacy of selection on braincase shape over any potential ecological influence

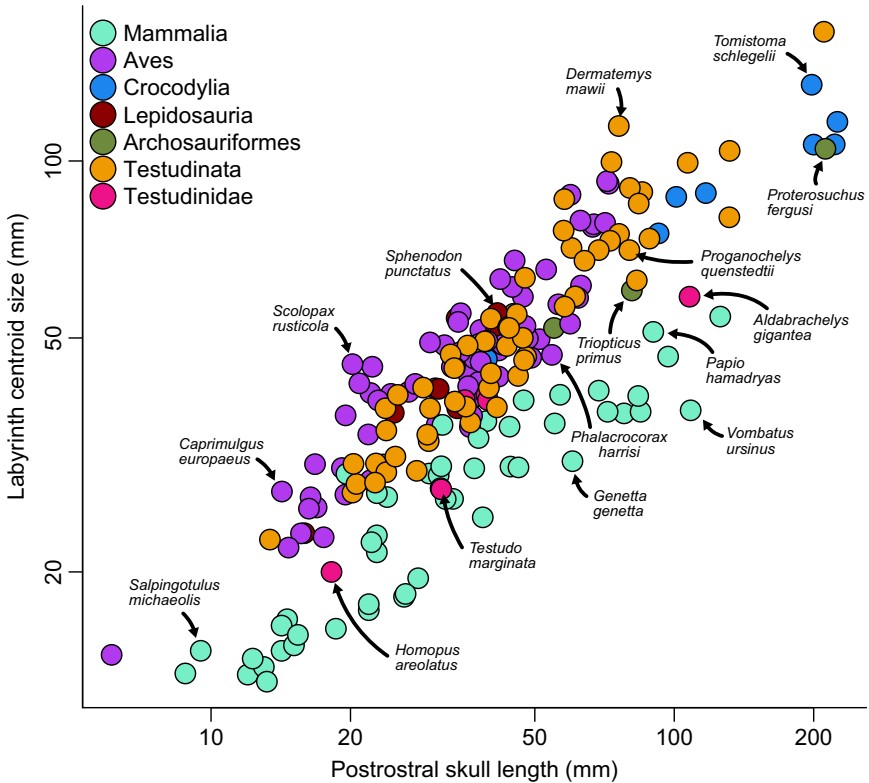

**Fig. 4 | Relative labyrinth sizes in tetrapods.** Labyrinth centroid size data plotted against postrostral skull lengths. A version of this figure which identifies each species is presented in Supplementary Fig. 10. Source data are provided with this paper.

on labyrinth shape specifically; and (3) similar or identical functional performance can result from highly different-shaped labyrinths[47], such that selection on functional performance of the labyrinth should not require very specific shape outcomes of the type that might over-rule selection on braincase morphology.

For these reasons, we advocate that future studies should explicitly compare both locomotor and non-locomotor explanations rather than limiting themselves to the former. More widely, weak or absent correlations of labyrinth shape with locomotion may also be explained by the recent observation that shape analysis, although widespread in the study of labyrinth form, provides an unreliable index of duct function[47]. Differently shaped labyrinths can have similar or identical functional performance, and same-shaped labyrinths can have different functional performance (depending on size); therefore, quantifying function directly from fluid flow models may provide a more fruitful direction for future studies[47].

We find that turtles, in general, have proportionally large labyrinths, contrasting with previous suggestions of small labyrinth size in turtles[24], and challenging the generality of functional interpretations of labyrinth size in tetrapods. This result contrasts with the hypothesis that large labyrinth sizes in tetrapods are related to high agility, or turning performance, in birds and some mammals[11,13,14,18,24]. Indeed, the relative size of the labyrinth in turtles exceeds that seen in mammals and rivals that of many birds (Fig. 4), but turtles are conspicuously less agile than those groups, and even some other reptiles. Our findings may be consistent with some previous observations that challenged the generality of a link between labyrinth size and head rotation. For instance, extreme labyrinth size reduction in cetaceans has been interpreted under the agility hypothesis as a reduced sensitivity counteracting increased body rotations experienced during diving[10] or due to the development of stiffened necks[26]. However, cetaceans do not experience stronger head rotations compared to terrestrial quadrupedal mammals[50]. Thus, functional interpretation of labyrinth sizes in cetaceans is challenging and may be not related to locomotor

ecology, but instead related to decreased importance of vision in cetaceans[50], consistent with an association of relatively large labyrinth size with eye size and visual acuity in mammals[25].

It also seems unlikely that labyrinth size variation among turtle species is related to agility, contrasting with findings about its linkage to variation in labyrinth size among mammals (e.g., refs. 11,14,44). The lack of quantitative agility data and defensible qualitative agility divisions in turtles makes this difficult to explicitly test. However, the turtle with the largest relative labyrinth size, the Central American river turtle *Dermatemys mawii*, is a slow-moving aquatic herbivore that nocturnally grazes on aquatic plants[51], making it a poor fit for the 'agile' category. In addition, our relative neck length variables are a proxy for head movability, but neither variation in neck length, nor variation in the neck retraction mode, nor the full vs. incomplete ability to retract the head explain variation in turtle labyrinth size, contrary to speculations of the previous studies[39]. Variation in relative internal canal diameter is large among turtles[52] (Fig. 1), and the internal semicircular duct diameter influences labyrinth response time[5] independent of duct length or labyrinth size. However, preliminary comparisons of membranous duct diameters to SCC diameters suggest that thick bony canals of sea turtles do not correlate with enlarged duct diameters, but instead accommodate an increased perilymphatic space (Supplementary Fig. 11). Therefore, variation in bony canal thickness most likely does not have a direct functional relationship to labyrinth response time.

We find a statistical association of large labyrinth size with aquatic habits, and of small labyrinths size with terrestrial habits among extant turtles. Our fossil data are also consistent with this association. For example, small labyrinth sizes were present in extinct turtles that independently evolved terrestrial habits, such as the Pleistocene meiolaniid stem turtle *Meiolania platyceps*, the Late Cretaceous stem-trionychian *Basilemys gaffneyi* and in Triassic–Early Jurassic terrestrial stem turtles, such as *Proganochelys quenstedtii*. Turtle middle ears interestingly show the inverse size relation, with middle ear cavities of

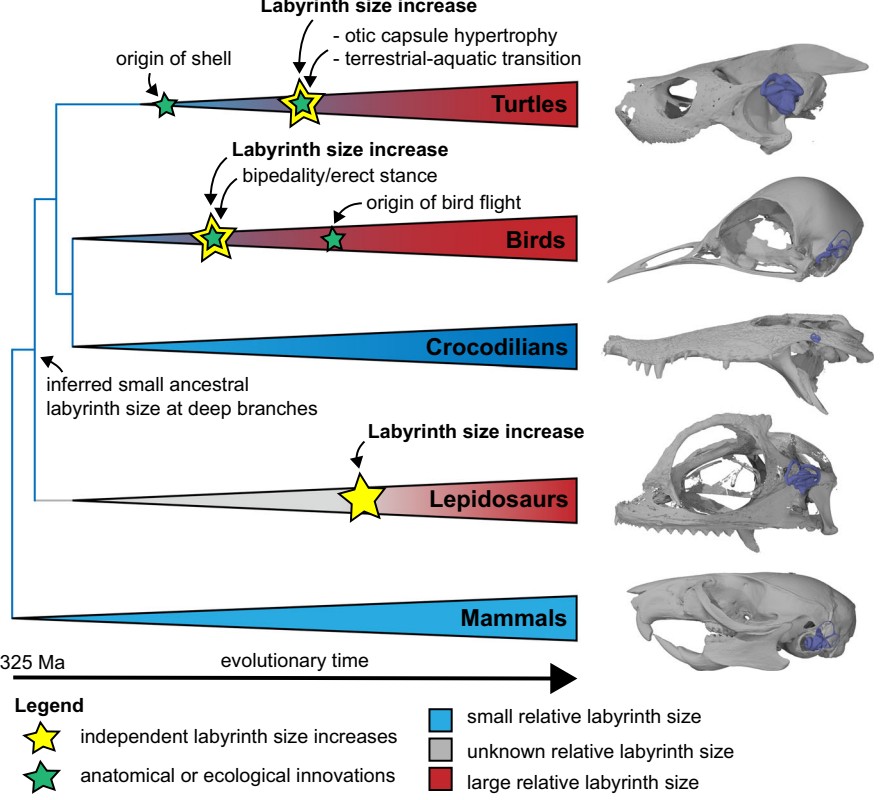

**Fig. 5 | Evolution of labyrinth size in amniotes.** Amniote phylogeny showing changes in relative labyrinth size across lineages. Labyrinth size increases at least three times independently, in the total-groups of turtles, birds, and lepidosaurs. The turtle relative labyrinth increase coincides with anatomical restructuring of the otic capsule and the ecological invasion of aquatic habitats. Images at tree terminals are cranial models with labyrinth models superimposed to illustrate relative labyrinth size, from top to bottom: *Dermatemys mawii* (SMF 564639); *Menura novaehollandiae* (FMNH 336751); *Crocodylus* sp. (BRSUG 28959); *Lyriocephalus scutatus* (OUMNH 1298); *Acomys cahirinus* (NHMUK 65.789.793). "Ma" is "million years before present".

extant aquatic turtles being smaller than those of terrestrial turtles[42], suggesting there may be a trade-off between labyrinth size and middle ear cavity volume.

Marine adaptation does not correlate with a reduction of labyrinth size in turtles, across multiple independent origins of marine habits in living and extinct groups. This contrasts with previous observations of marine, and especially pelagic reptiles and mammals with reduced labyrinth sizes, such as cetaceans[10], eusauropterygians[43], or metriorhynchids[48]. Our observations on marine, as well as aquatic turtle labyrinth sizes more generally, contrast with the hypothesis that secondarily aquatic tetrapods typically have small labyrinths compared to terrestrial relatives[10,44,48,53].

Our fossil data demonstrate that the reptilian ancestor had a proportionally small labyrinth (see also ref. 34) that was retained by early stem turtles, and that the evolution of large labyrinth sizes in turtles occurred independently to those on the bird stem lineage and within lepidosaurs (Fig. 5), most likely for different reasons. Increases on the line leading to birds occurred among flightless bipedal dinosauriforms, refuting the hypothesized association of large labyrinth size with flight[34]. Instead, increased visual acuity or bipedality in stem birds may explain the avian labyrinth size increase[34]. In contrast, the labyrinth size increase along the turtle stem lineage coincides with the evolutionary diversification of turtles into aquatic habitats, as well as important modifications in the skull, especially the otic region[38,42,54–56] (Fig. 5). Given the association of aquatic habits with large labyrinths in extant turtles, it is plausible that the evolution of aquatic habits explains the origin of large labyrinth size, although the functional relevance of large labyrinths in an aquatic environment remains unknown. It is noteworthy that aquatic turtles are not the only aquatic vertebrates with large labyrinth sizes, as these are also found in teleost

fishes[24]. However, this association is not generally present among vertebrates because many other aquatic taxa have proportionally small labyrinths (e.g., cetaceans[10]; sirenians and seals[44,53]) or intermediate-sized labyrinths (crocodilians[34]).

In freshwater turtles, active prey capture involves complex head-neck coordination[57,58]. In addition, visual cues have been determined as important for sub-aqueous prey acquisition in marine turtles[59]. This hypothesis is consistent with statistical evidence for a link between visual acuity and labyrinth size in mammals[25], and with the occurrence of large labyrinths in birds, especially in visual specialists such as raptors and albatrosses[32,34]. Therefore, we speculate that the large labyrinth size in turtles reflects an increased utility of visual field stabilization and coordinated head movements during prey acquisition in aquatic habitats. However, this and other explanations require further investigation. Possibly, the study of teleost fishes can be insightful, as teleosts heavily rely on vision[60] and have extremely large labyrinths[24].

The evolutionary increase in labyrinth size shown here on the turtle stem lineage may also be a driver of the rearrangement of the otic capsule and jaw adductor system, which are important traits of crown-group turtles[54]. Enlargement of the otic capsule resulted in the evolution of the unique trochlear pulley system of adductor jaw musculature[45,46,54–56,61–63]. Current models explain the evolution of this trochlear mechanism by an interplay of neck and skull modifications related to neck retraction of turtles[45,56]. However, drivers of the initial evolutionary enlargement of otic capsule size, which then necessitated a rearrangement of the jaw adductors, have remained unexplained. We propose that increase in labyrinth size at the base of Perichelydia may be the unknown factor causing otic capsule enlargement.

Overall, our results imply greater complexity in labyrinth shape and size evolution and ecomorphology than implied by some recent

comparative studies (e.g., ref. 35; but see refs. 6, 27, for a more nuanced perspective). Current functional hypotheses related to agility perform poorly in explaining size and shape variation in turtles. Moreover, labyrinth size increases in stem birds and stem turtles cannot be explained by an overarching functional hypothesis, indicating lineage-specific drivers for labyrinth size evolution. Soft-tissue variation in the structure and function of the vestibular organ might provide an explanation of this and requires further examination using functional models. For now, our results demonstrate that patterns of labyrinth evolution among vertebrates have been incompletely characterized, despite the high research interest in the morphology and function of this system. Future studies are needed to formulate alternative functional hypotheses for labyrinth size and shape variation and by increasing the systematic coverage among extant and extinct tetrapod groups as well as fishes.

## Methods

### Turtle dataset
Our turtle dataset includes 168 specimens, representing 90 extant and 53 extinct species (Supplementary Data 1–2). Our sample covers all major extant and extinct groups of turtles, including various lineages of stem turtles, and one-quarter of extant turtle species diversity[64]. We segmented endosseous labyrinths and skull models from CT data using Mimics v.15–19.0. 16 of our specimens were wet specimens subjected to contrast-enhancing staining techniques prior to scanning (see Supplementary Data 2 for specimen details), for which we segmented both the membranous and endosseous labyrinth. Membranous labyrinths were used for validation of our landmarking procedure (see below), but were otherwise excluded from analyses presented herein. CT data availability is detailed in Supplementary Data 1 and 3D models generated for this study were uploaded to MorphoSource (www.morphosource.org/projects/000372533).

### Labyrinth landmarks
We quantified the geometry of the vestibular organ using 3D landmarks and semilandmarks. We landmarked midline skeletons of isolated SCCs, which model the extent of the membranous duct from the endosseous labyrinth model based on generalized duct-to-canal relationships (see also ref. 34). Such reconstructions are necessary due to a poor correspondence between the shape of endosseous labyrinth and membranous labyrinth in turtles[52]. Midline skeletons are single-lines of voxels derived by shrinking the canal circumference from the exterior, and represent the mean endolymph flow path through a semicircular canal[7]. The skeletons capture the relative lengths, orientations and positions of the SCCs. Reconstruction of isolated semicircular canals, skeletonization via the autoskeleton function, and landmarking was performed in Avizo lite 9.2. Midline skeletons of the semicircular canals were landmarked using six conventional fixed landmarks, which describe canal intersections with their respective ampulla and the common crus (Supplementary Fig. 1). Open semilandmark curves were placed on the midline skeleton between these landmarks (Supplementary Fig. 1). Additionally, a closed semilandmark loop was placed around the inner perimeter of the anterior semicircular canal (ASC) to capture canal thickness and potential vestibular variation related to utricular and saccular modifications (Supplementary Fig. 1). Our landmarks quantify labyrinth shape aspects (duct lengths, mean flowpath, canal diameter) that are related to vestibular function, and have thus better justification to test vestibular ecomorphology than single point-landmarks on external endosseous surfaces that lack uniquely-relocatable point-features, contrary to claims by ref. 35.

### Validation of landmarking concept
We validated our SCC reconstruction and landmarking concept by visual comparisons of midline skeletons derived from membranous and endosseous labyrinths (Supplementary Fig. 1), and by testing for phylogenetic morphological covariation between landmarked membranous and endosseous labyrinths from the same set of 16 specimens for which we had stained CT scans (Supplementary Data 9). We separately landmarked SCC reconstructions and soft-tissue semicircular ducts and subjected both datasets to General Procrustes Analysis (GPA)[65] using the gpagen function of geomorph version 3.1.0[66]. This, and other morphometric and statistical analyses were performed in R version 3.6.0[67]. Procrustes shape coordinates for both types of data were then subjected to phylogenetic two-block partial least squares regression (2B-PLS)[68,69] as implemented in the phylo.integration function of geomorph. Skeletons of membranous ducts and endosseous canal reconstructions match closely (Supplementary Fig. 1). Covariation was significant and very high (r.gls = 0.96, $p = 0.001$, based on 1000 iterations), indicating that our endosseous labyrinth-based reconstructions reflect variation in the membranous labyrinth (Supplementary Data 9), justifying downstream analyses of vestibular shape based on landmarks derived from endosseous labyrinths.

### 3D geometric morphometrics
Landmark data of endosseous labyrinths (Supplementary Data 3) were subjected to GPA[65], again with the gpagen function of geomorph version 3.1.0[66], removing variation in centroid size, initial position, or orientation resulting in Procrustes coordinates that reflect shape differences among specimens[70]. Semilandmarks were allowed to slide along their tangent vectors during GPA, minimizing bending energy differences from the mean shape due to the initially arbitrary (equal) spacing of points[71–74].

We used Principal Component Analysis (PCA) of the Procrustes coordinates to evaluate the major aspects of shape variation across our full sample of specimens for endosseous labyrinths ($N = 168$; Supplementary Data 10). We used Procrustes coordinates as labyrinth shape data, and centroid size as labyrinth size data for analyses. Centroid sizes were extracted from the output object of the gpagen function, and serve as a shape-independent measure of labyrinth size[75].

### Phylogenetic framework
For the phylogenetic framework for statistical tests, we used the molecular phylogeny of ref. 76. For analyses including fossil taxa, we used a composite phylogeny in which fossil taxa were included into the molecular topology of extant turtles based on morphological studies (see Supplementary Methods for details on the phylogenetic topology). We time-scaled the tree based on molecular clock and fossil clade-age constraints using commands from paleotree v.3.3.0[77], Claddis[78], and ape[79]. Previously published divergence time estimates for crown clades[76] and fossil pleurodires[80] were used as minimum node age constraints. Ages of other nodes were calibrated using fossil age data (Supplementary Data 6) with the stochastic cal3 method[81] (Supplementary Data 7) and alternatively determined using a minimum branch length (mbl) argument that sets zero-length branches to a minimum of 1 Ma[77,82] (Supplementary Data 8). The effect of using different calibration methods was minor in phylogenetic comparative analyses (see Supplementary Tables 2–3). The analyses presented in the main text are based on the cal3-calibration. Time-scaled composite trees are shown in Supplementary Figs 2–3 and provided as Supplementary Data 7–8, and age data used for calibration are listed in Supplementary Data 6.

### Ecological variables
For phylogenetic comparative hypothesis tests, we defined ecological predictor variables that are related to habitat and locomotor ecology of extant turtles. We coded binary categorical variables (i.e., true/false) for locomotor behavior (burrowing, terrestrial walking, aquatic bottom walking, open water swimming) and habitat preferences (terrestrial, freshwater, marine) by reference to the literature (e.g., ref. 36).

Habitat preferences were also scored for fossil turtles when information was available in the respective literature. In addition, we categorized the degree of forelimb webbing (absent | minor | moderate | extensive | flippered) following the approach of ref. 42. Forelimb webbing is directly related to swimming kinematics[82,83] and thus serves as a proxy for aquatic adaptation[42,84–86], with a finer resolution than habitat preferences or main locomotor behavior. Forelimb webbing thus may reflect ecology more accurately than observational field data that underlies the other categorization approaches. Turtle necks facilitate head movements against the rigid shelled body and are related to head motion and feeding behavior, with long-necked species performing more rapid head movements[87,88]. Different anatomical configurations of the neck in stem turtles, pleurodires, and cryptodires additionally affect the ability for neck retraction as well as the primary plane of neck movements[46,63]. We used binary categorical variables as discretized neck length relative to carapace length categories (extreme > 70% neck-to-carapace length proportion, long = 50–69%, intermediate = 35–49%, short = <35%; Supplementary Table 1), and binary categorical variables to encode neck retraction ability (full, incomplete ability) and the type of neck retraction (none, horizontal, vertical). Specific explanatory variables for each species are given in Supplementary Data 2, and variables are further discussed in the Supplementary Methods.

## Labyrinth shape regressions

We evaluated the relationship of labyrinth shape (Procrustes coordinates) to labyrinth size, skull size, habitat ecology, and neck retraction ability and type using our dataset including fossils ($N = 138$) and Procrustes distance phylogenetic regression (procD.pgls)[89] based on the procD.pgls function of geomorph (Supplementary Data 11). For species with duplicate specimens, the largest individuals were chosen here and in all other statistical analyses. Information Theory model comparisons are currently not available for procD.pgls, due to problems with calculating the likelihood of the data given the model when the number of traits (= landmarks) exceeds the number of specimens[90]. Therefore, we informally compared procD.pgls regression models based on their coefficients of determination ($R^2$), the significance ($p$-values) of variables, and effect sizes ($Z$-scores). Variable significance and effect sizes are calculated from (Type II) sums-of-squares of the phylogenetically-transformed Procrustes distances and sum-of-squares distributions generated from residual randomization permutation procedure (RRPP)[91] using 1000 iterations. $P$-values were not adjusted for multiple comparison, following recent literature using the same procedure[34,92]. Effect sizes indicate the strength of relationship between shape as the response variable and the explanatory variable, and can be compared among regressions[93,94]. We employed an iterative process of model evaluation, whereby significant variables of initial bivariate model tests were carried forward to more complex models including combinations of multiple explanatory variables. We tested allometric effects on labyrinth shape using models of the formula (shape ~ size), whereby size variables were $\log_{10}$-transformed linear skull measurements, skull box volume, and labyrinth centroid size. We also tested whether spatial constraints in form of the aspect braincase ratio (skull height/skull width) had an effect on labyrinth shape, which may be expected as turtles show considerable variation in their relative skull flatness[95]. Ecological effects on labyrinth shape were tested in a series of models that considered all variables individually (shape ~ ecological variable) and those that additionally accounted for more complex effects (e.g., shape ~ skull size × labyrinth size + ecological variable). Interaction terms between allometric effects, as well as between allometric effects and spatial constraints were also tested. We also conducted additional versions of these analyses: (i) excluding all marine taxa, (ii) excluding chelonioid sea turtles, and (iii) omitting the landmark loop around the inner ASC perimeter (Supplementary Notes; Supplementary Data 11).

We visualized the effect of skull size and braincase aspect ratio by plotting their regression scores against each predictor (i.e., skull size and braincase aspect ratio, respectively). We also visualized the shape deformations associated with variation in skull size and aspect ratio. This was done by multiplying the regression coefficients for each predictor by their minimum and maximum values whilst holding the other predictor of the multiple regression at its mean value, resulting in a predicted labyrinth shape for extremes (i.e., minimum, maximum) of each predictor. The differences between minimum–maximum pairs were normalized and colored-coded to display the gradation of change in landmark position.

## Size corrected labyrinth shape analyses

Because shape regressions indicated a significant effect of skull size and the braincase aspect ratio on labyrinth shape (see results), we performed a second PCA on the residuals of a regression of the form 'shape ~ skull box volume + braincase aspect ratio' (Supplementary Data 12). This PCA represents the labyrinth shape space once corrected for allometric shape variation and for shape variation imposed by changes to braincase proportions due to spatial constraints. Because the phylogenetic regression can only consider one specimen per species (for which we used the largest available specimen), the corrected PCA contains fewer datapoints ($N = 138$) than the uncorrected PCA. This also excludes five species for which we had no complete skull size data.

## Labyrinth size regressions

We used multiple pGLS[96] regressions to test statistical associations of labyrinth size with explanatory variables representing the effects of allometry (head size), spatial constraints (braincase aspect ratio), and locomotor and habitat ecological traits (as described above) on our extant-only dataset ($N = 89$) for which all ecological data were available, and on a smaller dataset ($N = 56$), which also included relative neck length data (Supplementary Data 13). Here and elsewhere, all continuous valued traits were $\log_{10}$-transformed prior to analysis. We used combinations of these variables in multiple regression models to evaluate the independent effects of skull size, aquatic adaptation, and other ecological traits on labyrinth size. Regression models were compared using AICc[41], implemented using the AICc function from qpcR 1.4.1[97]. We computed AICc weights[98,99], discarding models with less than 1/10th the weight of the best model as negligible[99]. For all pGLS regressions, we calculated the generalized coefficient of determination ($R^2$) from the maximum likelihood values of any given model and those of an intercept-only null model following the equation of ref. 100. Coefficients of explanatory variables were estimated using pGLS restricted maximum likelihood. The $t$-statistics are coefficient estimates divided by their standard error. $P$ values are two-sided, and are calculated using the coefficient value and a $t$-distribution with the number of residual degrees of freedom (re-df) of the model.

## Evolution of turtle labyrinth size

We visualized the evolution of relative labyrinth size using phylogenetic optimization. Proportional labyrinth size for 138 extant and extinct turtle species was characterized as the predicted labyrinth size residuals from the AICc-best performing pGLS regression. Measurements were $\log_{10}$-transformed prior to analysis. pGLS was implemented using the gls function from package nlme version 3.1.141[101]. These analyses used the largest specimen of each species where multiple specimens were available. The phylogenetic covariance structure was considered using the corPagel argument from ape version 5.0[79], estimating the phylogenetic signal parameter lambda[102] during the estimation of regression parameters. Ancestral state estimation was conducted using the ace function of ape on pGLS residuals rescaled between 0 and 1 (Supplementary Data 16).

## Comparative labyrinth size among tetrapods

To evaluate the relative size of the labyrinth in turtles compared to other amniotes, we compared labyrinth centroid sizes with postrostral skull lengths of 200 tetrapods, including five extinct archosauriformes, eight extant crocodylians, seven extant lepidosaurs, 50 extant mammals, 65 extant birds, 61 extant turtles and 4 extinct stem turtles (Supplementary Data 18). Archosauriform and crocodylian landmark data were taken from ref. 34, bird data from ref. 32, whereas the mammal and lepidosaur data are published within this work (see Supplementary Data 1 and 18). All landmark datasets were combined and inspected, and measurement units were adjusted to mm for all specimens across the datasets. Centroid sizes were calculated from a joint PCA of all landmarked labyrinths (Supplementary Data 20). Postrostral skull length as a body size proxy avoids effects caused by the elongated rostra or beaks of many birds or crocodiles (see also ref. 103. as an example of excluding rostral effects in mammals), and could be measured from the same specimens from which the labyrinths were segmented. Although rostrum length may affect the rotational moment of inertia for skull rotations and could thus have relevance to labyrinth function, empirical evidence shows that full skull length correlates less with labyrinth size than postrostral skull length in birds and crocodiles[34].

## Reporting summary

Further information on research design is available in the Nature Research Reporting Summary linked to this article.

## Data availability

All CT data gathered for this study were deposited in MorphoSource. All 3D models were deposited in MorphoSource, and can be accessed here: www.morphosource.org/projects/000372533. Supplementary Data 1 list webpage links to the labyrinth models of all used amniote species to facilitate download. The parent CT scans for each model are directly linked with the deposited 3D model in MorphoSource, with a few exceptions when CT data were previously deposited by other authors in different repositories. The CT scan availability for scans not currently housed in MorphoSource, including information about restricted download policies implemented by many museums within MorphoSource, is detailed in Supplementary Data 1. Supplementary Text (including supplementary Tables and Figures) and Supplementary Data files were uploaded directly to the journal. Supplementary Data files are also available on GitHub at (https://github.com/SerjoschaEvers/Turtle-Labyrinth-Ecomorphology-and-Evolution-Data), with the version published in this paper available on Zenodo at (https://doi.org/10.5281/zenodo.7024572)[104]. Source data are provided with this paper.

## Code availability

All R codes (Supplementary Data 9–13, 16, 20) and raw data files (Supplementary Data 2–8, 17–19) required to reproduce the analyses are available as Supplementary Data files directly uploaded with the journal and additionally published on GitHub at (https://github.com/SerjoschaEvers/Turtle-Labyrinth-Ecomorphology-and-Evolution-Data). The version published in this paper is available on Zenodo at https://doi.org/10.5281/zenodo.7024572[104].

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

## Acknowledgements

We thank Farah Ahmed, Jérémy Anquetin, Paul Barett, Don Brinkman, Patrick Campbell, Kristian Carlson, Nathan Carroll, Sandra Chapman, Matt Colbert, Loïc Costeur, Cécilian Cousin, Tom Davies, Annelise Folie, Jason Head, Tea Jashashvili, Marc Jones, Hilary Ketchum, Alexander Kupfer, Stephan Lautenschlager, Joshua Lively, Zhe-Xi Luo, Tyler Lyson, Jessica Maisano, Carl Mehling, Ben Moon, Timothy Myers, April Isch Neander, James Neenan, Mark Norell, Alex Peaker, Oliver Rauhut, Alan Resatar, Matt Riley, Yann Rollot, Tim Rowe, Bruce Rubridge, Torsten Scheyer, Rainer Schoch, Anne Schulp, and William Simpson for facilitating data collection and/or sharing data. This research was funded by a National Environment Research Council studentship on the Doctoral Training Partnership Environmental Research number NE/L0021612 (to S.W.E.); a Swiss National Science Foundation Ambizione Fellowship number PZ00P2_202019/1 (to S.W.E.); Fundação de Amparo à Pesquisa do Estado de São Paulo grants 2016/03373-0 and 2019/02086-6 (to G.H.); the International Partnership Program of Chinese Academy of Sciences, Grant No. 132311KYSB20190010 (to J.N.C. and H.Y.); a German Academic Exchange Service grant 91546784 (to C.F.); and Swiss National Science Foundation grants 200021_156087 and 200021_178780/1 (to W.G.J.).

## Author contributions

S.W.E., W.G.J., and R.B.J.B. designed the research, S.W.E., W.G.J., J.N.C., G.H., H.Y., I.W., R.B.J.B. collected scan and model data, S.W.E. and C.M.J.

landmarked the data, S.W.E., G.H. and R.B.J.B. performed analyses, S.W.E. made figures, S.W.E. and R.B.J.B. wrote the paper, W.G.J., J.N.C., G.S.F., C.F., G.H., H.Y., C.M.J., I.W. contributed to the text.

## Competing interests

The authors declare no competing interests.
