## [Peer Review File · Nature Communications]

Independent origin of large labyrinth size in turtlesReviewers' Comments:

Reviewer #1:

Remarks to the Author:

This study is a very wide and meticulous morphometric survey on turtles' labyrinth. It involves considerations, notions, methods and principles in comparative anatomy, shape analysis, ecology, and phylogeny, and it deals with reptiles as well as with birds and mammals, living and fossils species. The outstanding wide view is a strength of this paper and, at the same time, it is probably its main weakness. The dimension of the study is so vast that many results and conclusions (or their implications) may easily pass unappreciated. At the same time, the large number of methods, techniques, concepts and assumptions make the paper very hard to revise in detail. Despite the massive supplementary data (49 pages!), many methods or criteria may still look scarcely supported or documented, when compared with more standard articles. Similarly, because of the many distinct issues and hypotheses discussed in the study, real conclusions are apparently mixed with speculations and personal opinions. I really appreciate this study, and the contents look competent and balanced. But I strongly suggest splitting it into two or even three distinct papers. At first glance, I can suggest the following ones: i) a methodological article, to acknowledge all the (in some cases, subjective) complicated morphometric passages and decisions; ii) one dedicated to turtle (within-group) variation, dealing with species, families and ecology; and iii) one dedicate to the comparison with other groups (reptiles, birds, mammals), dealing with vertebrate evolution.

Please find below further comments on specific issues:

1. As far as I can understand, you suggest that locomotion/ecology is not much relevant when dealing with labyrinth morphology. However, the PCA apparently shows the opposite, because it looks like the three groups, albeit overlapping, do show distinct distributions.
2. In the results/discussion section, I suggest separating more clearly the parts on size and the parts on shape. At present, I have some difficulty in following the different results.
3. You use the braincase proportions (aspect ratio) as proxy for spatial constraints. This is a reasonable approach, but probably, in the discussion, you can step a bit further into this topic, speculating on possible specific factors associated with turtles' temporal anatomy.
4. Do you think this morphological variation of the labyrinth has something to do with the "anapsid-like" condition? A note on this point may be interesting.
5. I strongly recommend increasing the size of the figures. Figure 1c, 1d, 2b, 2c are really hard to read. In figure 2a and 2b, I understand that the colour deals with size, but this has to be probably better explained. All the figures are very good ones, a very good job indeed.

Congratulations for such extensive and detailed study.

Reviewer #2:

Remarks to the Author:

This interesting paper provides evidence that the inner ear labyrinth of turtles is larger in size than would be predicted by current thought on the function of semicircular canals, which frankly has been somewhat limited to mammalian studies. The expansion of our knowledge on this front into different amniote groups is exceptional, and the paper is particularly impressive in the amount of data analyzed and effective use of different sorts of morphological analyses including phylogenetically-informed analyses. I think the paper is in great shape for publication – as I was reading one part of the discussion section I started thinking about whales (and how I want to apply the same method to them

– have been thinking about testing this eye size to labyrinth size question) and then the authors began to discuss them, so I was especially excited to see that. I believe this will lead to other larger studies and bump the field into new directions. All of the methods seem sound – I was not quite clear how the centroid size of the labyrinths, to be used as a measure of size, was calculated. If a few sentences could be added explaining that, plus a few minor edits perhaps from my annotated documents, I think the paper will be ready for publication.

With best regards,
Rachel Racicot, PhD

Reviewer #3:

Remarks to the Author:

This is a great project, everyone! By and large this manuscript is in really good shape. Most of the issues that I see are in the background info where more clarity is required:

- 1) Go through all the references carefully. I don't know all of them by heart, but I do many of them and find that there are quite a few that are inaccurately applied/attributed.
- 2) Your data are awesome enough on their own you don't need to continually over-emphasize the pervasiveness and legitimacy of some "aquatic turtles have small canals" hypothesis, which really doesn't formally exist.
- 3) You occasionally slip into the same contradictions as Spoor et al. both in terminology and interpretation and this has a few significant impacts

I present for your consideration also, a few additional points that should be addressed either by further analysis (in some cases easy, in some admittedly not), or at least direct address in the ms.

Specific comments:

In 20: "predictable effects on sensitivity". 'Sensitivity' here and throughout is always a problematic choice of words, even if it follows that of earlier works. In my view, many of the canal hypotheses that have been falsified over the years suffered because the idea of combined 'sensitivity' of the canal is not meaningful. By the time this ms gets to the end, it teeters on the same precipice by over-emphasizing some perceived magic meaningful connection between canals "size" and "sensitivity" in the comparison between turtles and birds. Canal response/efficacy is a matter of both response time and signal discrimination and changes in some parameters can have opposite effects on those two functions. Is a canal more "sensitive" if it's discriminatory capability goes up but it's response time decreases? I realize you don't have the space in this ms to go into all the biophysics, but the introduction and the interpretation of the results will be improved if you make this idea more clear.

In 18: "reflexes that stabilize the visual field" Careful about overstating the role of the canals in visual stabilization. Vestibular signals are important primarily because of their low latency, not because they are the perfect solution to the problem of visual stability. For truly visual species especially, opto-kinetic signals are the real workhorses here, but they are slow enough that the fast vestibular reflexes get things moving in approximately the correct direction first. There's no question VOR and VCR are important, but if they are not the ultimate arbiters of a stable field of view then perhaps moderate to low correlations between vestibular morphology and locomotion are not too surprising.

In 37: I disagree with the choice of Spoor and Zonneveld to exemplify the correlation between canals and locomotion. 1) The only locomotor correlation they claim to demonstrate is human vs all other primates, 2) the actual statistical support for this is dodgy at best, 3) they outright state in their conclusions that this correlation is not strongly supported, 4) they further state in their conclusions that there are many other more important factors in the morphology of human canals than the flimsy locomotion connection.

In 38: I've got no complaint about presenting Spoor's agility-based data (it will always remain an important historical step in this field), but as with his use of sensitivity, 'agility' is not a very meaningful biomechanical term (as you cover in the discussion). It's clear from the cited works and conversations with him that he is trying for something akin to turning-performance as a metric (hence his insistence on body mass as well, as a determinant of the rotational moment of an entire animal turning).

In 39: Part of your investigation later on is relative canal size, so I strongly recommend against presenting Kemp and Kirk (2014) without some further commentary on the weakness of their conclusions relative to size. I know Roger has used this in his 2017 paper, and, yes, Kemp and Kirk state that eye-size is a primary driver of canal size but my concern is that if you look at their "eye-size" variable, it is residuals from an eye/body mass regression. Without any direct evidence that eye-size is independent of the overall size of the head (unlikely - so there's also the autocorrelation issue here), what they are interpreting as eye-size to support their visual argument is, based on their data, not actually distinguishable from a simple head-size argument which may or may not favor a biomechanical interpretation that directly links canal size to head motion without reference to the function of the visual system.

In 50: The three papers cited other than Benson et al all find statistical support of one form or another for locomotor correlation and Hanson et al in some ways directly contradicts Roger's earlier work. Does this really demonstrate "little statistical support"?

In 64: I see no references for these "studies", but any study that suggests turtles do or should have small canals should be taken with several very large grains of salt as it indicates some very fundamental misunderstanding of turtle canal morphology and basic vestibular principles. If you mean that this is indirectly implied by studies which follow the Spoor model of ponderous locomotion = small canals, then please rethink this, as the different claims of size-to-"agility" relationships in Spoor's model are not even internally consistent (positive general correlation means bigger/less "agile" animals have larger canals...but somehow positive residual on same model equates to more "agile" animal? This has never been adequately explained or defended) let alone particularly broadly applicable.

In 117: This is a completely inaccurate citation. Nowhere in that chapter do we claim that aquatic reptiles have overall small canals (absolute or relative); nor would I ever do so (see comment for In 64 above). There are, in fact, no claims of any kind about overall size of canal systems in that chapter as it is entirely based on size independent shape analysis. Actually, the very surprising thing is the small size in the testudines. Given the "free-fall" portion of their step-cycle as the CoM crosses the diagonal line of support (Zug 1971) I would expect larger canals (though the thickness of the canals might be sufficient adaptation here). Perhaps the idea that is being referenced here is merely the statement of eccentricity of the canal circuit in aquatic taxa where the area enclosed by the circuit is smaller *relative to the canal perimeter* than in terrestrial taxa a claim further supported by Goyens analysis.

In 120-124: This is fascinating. I admit I'm a little surprised by this finding, but only a little as I would expect the length of the neck to be far less relevant than the cross-sectional area of the neck. Neck length does represent some small contribution to the moment of inertia of the head/neck complex, but it's not capturing the capability of neck stabilizing muscles to reduce/counter rotations. I realize that muscle area is probably not a feasible parameter to collect but I would encourage the addition of some other proxy for neck muscle mass - perhaps average of (width * height) across the cervical vertebral series is possible on the specimens where you have a neck length value.

In 151-153: See comments in previous section. I don't believe the large size of turtle canals should be characterized as surprising at all, nor should any significant stock be placed in the assumption of small size based on slow locomotion.

In 169: I am curious what happens if you leave out the sea turtles. In the 2008 chapter we also found no significant shape difference within our entire turtle sample but, when we left out the chelonids due to their unusually derived vestibular morphology we uncovered a strong signal within the remaining turtles that converged on the signal within the other amniote groups. This is especially true in your dataset as I strongly doubt that the grade of forelimb webbing truly captures the magnitude of difference between chelonids and other turtles (Pace looked at a flippered trionychid). I suggest that the inclusion of sea turtles here is masking several more important findings. This should at least be discussed but better would be a separate, sea turtle free, analysis.

In 171: Again, many of these citations are inaccurate in this context:

- 1) In my 2008 work it explicitly states that "Canal shape is, to a large extent, phylogenetically conservative" and the locomotor adaptation is "secondary variation".
- 2) Malinzak et al. discuss primarily the angles between canals, a feature not even addressed in this ms so I'm not sure it's really fair to critique any assumptions they may or may not have made (and a primacy of locomotor related canal variation is certainly not an explicit assumption in the paper).
- 3) Our 2013 work only ever discusses "adaptive" changes and make no claims of the magnitude of these changes with respect to phylogenetic base-line shape.
- 4) Grohe et al. explicitly test, find, and state several times that phylogenetic signal in their mustelids is stronger than the locomotor signal.
- 5) Eric, to my recollection makes no such claim in his 2016 paper.
- 6) OK....possibly Goyens...

In 174: It is inaccurate to claim that Georgi et al. 2013 neglected phylogenetic information. Although we did run analyses on the raw data, all of the significant findings reported were PICs and even the figures are explicitly labeled as showing the regressions and CIs of phylogenetically corrected OLS.

In 177-179: Werner Graf was making this argument in response to some of the earliest work back in the 1980's but 40 years later solid data remain elusive. Your numbers tell a compelling story of correlation, but I wonder if your interpretation of the correlation is correct given the size range of the samples...see Goyens for example. I certainly don't think this interpretation has quite the strength it is presented with. In many cases, the flattening of the skull is likely an aquatic adaptation in itself so perhaps this just once again means that the webbing index is not as informative as hoped.

In 188 and on: I strongly suggest that many of these direct comparisons between turtles and other groups be qualified a little (ok, if it were me, a lot). Yes the relative size comparison is eye-catching but things get too dicey when you begin to compare turtles canal response directly to other animals, especially if you are just using one (less critical than several others) parameter. There are just too many weird aspects in the turtle system:

- 1) the papilla neglecta in turtles seems to encode some semicircular duct flow signals from the common crus (Brichta and Goldberg 1998), that's going to significantly alter neural response.
- 2) Nothing else that I have seen (other than possibly a few ankylosaurs) has the fat-sausage canals of some of these turtles. From a neural response standpoint, the potential for a large duct radius to swamp out nearly all meaningful variation in duct length is enormous.
- 3) Utricular and saccular modifications in some (many?) of these taxa may well have an impact on flow within the duct circuits, especially in light of the the extreme duct cross-sections. See comment for In 298-300

As in so many other areas, when turtles fail to fit a larger pattern the default assumption must be that it is turtles that are divergent not that the pattern itself is less applicable.

In 190-193: Your results do indeed contrast with the agility hypotheses, but this is not "unexpected" because those hypotheses are nearly worthless and already counterindicated by several other studies you have already cited up to this point.

In 227: Again, there is NO hypotheses of small size in aquatic animals in Georgi and Sipla 2008 either stated or implied.

In 239: Up to this point you have restricted the discussions to amniotes (perhaps because of the emphasis on the claim of small size in aquatic taxa, which is the ms biggest weakness). But if you're going to now postulate a relative large size due to an aquatic habitat you really should bring the fish and their HUGE canals up from the last line of the ms to a fuller discussion right here.

In 251-260: This either needs to be fleshed out a little more or dropped completely; as it is presented at the moment it (mostly) contradicts the earlier discussion that the canal morphology is constrained by the brain-case. That is a shape argument and this is a size argument so it is possible that you can find a congruence between the two, but currently it doesn't work (personally I think it's the brain-case argument that is the issue, see comment for In 177).

In 262: The "complexity in labyrinth shape and size evolution" has been recognized since Gray's work at the turn of the 20th century, so I think this needs some clarification.

In 264-266: That's because "size" alone is an awful correlate of actual functional response in a broad taxonomic comparison and really shouldn't ever be used to compare birds and turtles (see comments for In 188). I am 0% surprised that this comparison appears meaningless.

In 298-300: I'm concerned about this closed loop of the internal ASC perimeter metric. Many of your specimens, rather than showing extreme morphology of the canals, demonstrate extreme morphology of the vestibule itself driven by utricular and saccular modifications (chelus fimbriatus springs to mind particularly, but it seems you've got some Gopherus and a few other culprits as well). I suspect that this metric is capturing not a little variation from these extremes and not as fully representative of canal morphology as you would like. The problem is bigger in the PSC and LSC, of course, but I don't think the ASC is free of it.

In 406-416: Yes, using postrostral skull length "avoids effects caused by the elongated rostra" but it also neglects what could possibly be some of the most functionally relevant morphology. How easily the head rotates, or conversely how stable the head is, is fundamentally based on the rotational moment of inertia and that, in turn, is driven by the distribution of mass to the second power, $\sigma (m * r^2)$, giving the rostral skull (the mass most distant from cervical rotations) considerably more functional relevance than the postrostral skull.

To my knowledge ontogenetic scaling of the optic capsule in turtles has not been quantified. However, in every other taxon where the bones of the otic capsule are not fused, there is post-natal growth of the inner ear structures. So much of the results and discussion of this work is focused on the relative size of the inner ear system some this issue should be addressed either in the specimen choice (i.e., are all the specimens at least roughly of mature size) or as an explicit discussion in the results section. I think this is especially important here since some of the larger taxa investigated see growth well over an order of magnitude in length alone.

Except for the few issues mentioned above, the material and methods look good.

Cheers All,
Justin

Point-by-point response:

Reviewer #1 (Remarks to the Author):

This study is a very wide and meticulous morphometric survey on turtles' labyrinth. It involves considerations, notions, methods and principles in comparative anatomy, shape analysis, ecology, and phylogeny, and it deals with reptiles as well as with birds and mammals, living and fossils species. The outstanding wide view is a strength of this paper and, at the same time, it is probably its main weakness. The dimension of the study is so vast that many results and conclusions (or their implications) may easily pass unappreciated. At the same time, the large number of methods, techniques, concepts and assumptions make the paper very hard to revise in detail. Despite the massive supplementary data (49 pages!), many methods or criteria may still look scarcely supported or documented, when compared with more standard articles. Similarly, because of the many distinct issues and hypotheses discussed in the study, real conclusions are apparently mixed with speculations and personal opinions. I really appreciate this study, and the contents look competent and balanced. But I strongly suggest splitting it into two or even three distinct papers. At first glance, I can suggest the following ones: i) a methodological article, to acknowledge all the (in some cases, subjective) complicated morphometric passages and decisions; ii) one dedicated to turtle (within-group) variation, dealing with species, families and ecology; and iii) one dedicate to the comparison with other groups (reptiles, birds, mammals), dealing with vertebrate evolution.

Response: We thank reviewer 1 for the kind words about our work. We also appreciate that the topics we cover within our manuscript may have been suited to be broken up into more papers, although we have already done that. For example, our landmarking method was presented already in last year in Bronzati et al (2021) (for which we did the landmarking & analyses), and we have tried to signal this more clearly in the revised MS. The necessary steps of our method are all documented in the main text and supplements, and analyses are documented in the text of the methods and supported by supplementary R scripts. We made a few minor modifications to the methods section that explain the landmarking concept and that justify our choice of landmark concept over alternatives, such as those used by Hansen et al. (2021). We also are planning a follow-up study, after the current paper, to explore the within turtle-group variation in depth and explore patterns of disparity between the various turtle clades, as well as an assessment of SSC thickness specifically. As the current paper deals primarily with labyrinth ecomorphology and evolution, we chose to include PCA plots colour-coded by ecology, but leave out taxonomic assessments of disparity for the anticipated follow-up study. We also think that the ecological evolution of turtles influences labyrinth size evolution. In agreement with the editorial team, we have thus decided to not further split this paper.

Please find below further comments on specific issues:

1. As far as I can understand, you suggest that locomotion/ecology is not much relevant when dealing with labyrinth morphology. However, the PCA apparently shows the opposite, because it looks like the three groups, albeit overlapping, do show distinct distributions.

Response: It is true that the colour-coded PCA data gives the impression of ecological separation of labyrinth shapes. However, all ecological classes have distinct overlap near the centroid of the morphospace. Furthermore, an apparent correlation between ecology and morphology in these plots could result from either (i) An actual correlation, or (ii) Shared phylogenetic signal in both shape and ecology, or (iii) Potential other explanations. It is impossible to eyeball the ‘true’ effect just from looking at the PCA. This is the reason why we provided statistical regression models that explicitly test between alternative explanations. We have explained this in the revised Results section as follows: “Therefore, the graphical association between locomotion and labyrinth shape in our morphospaces (Fig. 1) is not explained by direct functional linkage but by other factors (e.g. variation in soft tissue structure of the vestibular organ, braincase architecture, or other uncharacterised aspects that are embodied by ‘phylogenetic signal’ in our analysis).”.

2. In the results/discussion section, I suggest separating more clearly the parts on size and the parts on shape. At present, I have some difficulty in following the different results.

Response: In our previous MS version, shape and size results and discussions were separated by means of paragraphs. We now inserted subheadings (labyrinth shape variation in turtles; labyrinth shape regressions; labyrinth size regressions; labyrinth size evolution of turtles; amniote labyrinth sizes) to the results section, which is in line with the journal formatting guidelines. We hope this helps with better distinguishing the different topics discussed. In the discussion section, we use the same sequence (first shape, then size), but have no subheadings (in line with the journal style).

3. You use the braincase proportions (aspect ratio) as proxy for spatial constraints. This is a reasonable approach, but probably, in the discussion, you can step a bit further into this topic, speculating on possible specific factors associated with turtles’ temporal anatomy.

Response: We think that commenting on the temporal emarginations of turtles would require more sophisticated versions of capturing skull shape (landmarks, rather than an aspect ratio). This should for sure be explored in the future, although the temporal arrangements have been associated primarily with neck retraction in turtles (see Werneburg 2015), which we included into our analyses but didn’t find to be significant. We therefore think the documented relationship is more similar to recent data presented for squamates (Goyens 2019), in which skull flatness reduces available otic space causing labyrinths to be more flattened.

4. Do you think this morphological variation of the labyrinth has something to do with the “anapsid-like” condition? A note on this point may be interesting.

Response: This question is closely related to the previous one above. We do not think that the turtle-specific temporal arrangement is driving the observed pattern, especially as similar spatial constraint patterns have been documented for non-turtle reptiles (Benson et al. 2017; Goyens 2019; Bronzati et al. 2021), but without proper statistical tests, we cannot make decisive statements about this.

5. I strongly recommend increasing the size of the figures. Figure 1c, 1d, 2b, 2c are really hard to read. In figure 2a and 2b, I understand that the colour deals with size, but this has to be probably better explained. All the figures are very good ones, a very good job indeed.

Response: Thank you for these suggestions. We increased the label sizes in figures 1 & 2 by 1–2 point sizes, depending on the label type. We also increased the size of the colour legends in figure 1 by 120% and in figure 2 by 150%. We included a statement regarding the colours into the figure legend of figure 2 ("Colours denote relative labyrinth size and are consistent among all panels"). Thank you for the general praise of our figures.

Reviewer #2 (Remarks to the Author):

This interesting paper provides evidence that the inner ear labyrinth of turtles is larger in size than would be predicted by current thought on the function of semicircular canals, which frankly has been somewhat limited to mammalian studies. The expansion of our knowledge on this front into different amniote groups is exceptional, and the paper is particularly impressive in the amount of data analyzed and effective use of different sorts of morphological analyses including phylogenetically-informed analyses. I think the paper is in great shape for publication – as I was reading one part of the discussion section I started thinking about whales (and how I want to apply the same method to them – have been thinking about testing this eye size to labyrinth size question) and then the authors began to discuss them, so I was especially excited to see that. I believe this will lead to other larger studies and bump the field into new directions. All of the methods seem sound – I was not quite clear how the centroid size of the labyrinths, to be used as a measure of size, was calculated. If a few sentences could be added explaining that, plus a few minor edits perhaps from my annotated documents, I think the paper will be ready for publication.

With best regards,
Rachel Racicot, PhD

Response: We thank Dr. Racicot for their nice comments regarding our MS. Regarding the question of labyrinth centroid size: This measure can directly be extracted from the output of the geomorph Procrustes Analysis. Centroid size is defined as the square root of the sum of squared distances of all landmarks of an object from their centroid. Centroid size can be used to compare any shape described by a consistent number of landmarks in terms of their size. It basically allows a 'fair' comparison between sizes of shapes, as the shapes themselves can have effects on the sizes of non-centroid-size measures. For instance, the length of the labyrinth may be affected by the depth of the braincase, but centroid size is a size measure that is uncorrelated with shape (Bookstein 1986). We clarified the use of centroid size and how it was acquired with an additional sentence in the respective methods section: "Centroid sizes were extracted from the output object of the *gpagen* function, and serve as a shape-independent measure of labyrinth size (Bookstein 1986)".

We copied the comments from the annotated review file into this document to provide point-by point responses to Dr. Racicots comments:

l. 182: This could be why the semicircular canals are small in whales!!!

Response: Yes, this could indeed be true. We don't advocate this point here, because it would be speculative at this point, but hopefully future studies will address the small labyrinths of whales with respect to braincase constraints.

l. 202: Although, whales do have a large variety of eye sizes – this would need to be tested (it's been on my to-do list...). Also, I think a word is missing at the end of this sentence.

Response: This interesting. I don't know of any quantification efforts to eye size in whales, but good to know that it's on someone's to-do list! Thanks also for catching the missing word, we fixed the sentence.

l. 314: How is the centroid size converted to labyrinth size? I don't understand how a centroid has a size?

Response: See comment above. We now explain how centroid size was attained and why we use it (“Centroid sizes were extracted from the output object of the *gpagen* function, and serve as a shape-independent measure of labyrinth size (Bookstein 1986)”).

l. 395: I feel like this might be telling me how to get the sizes but it's still not completely clear to me.

Response: We hope that our previous comments clarified the use and nature of centroid sizes.

l. 416: We did this in the Racicot & Colbert 2014 (Anat Rec) to good effect!

Response: Thanks, I'm glad this also worked for you. Another reviewer also commented on the use of postrostral skull length (indicating it may omit important functional signal due to the effect of rostra on skull rotations. We now cite statistical evidence that postrostral skull length correlates better with labyrinth size in reptiles than full skull length does (Bronzati et al. 2021), and add the Racicot & Colbert (2013) citation as a case example for usage in mammals: “Postrostral skull length as a body size proxy avoids effects caused by the elongated rostra or beaks many birds or crocodiles (see also Racicot & Colbert 2013 as an example of excluding rostral effects in mammals), and could be measured from the same specimens from which the labyrinths were segmented. Although rostrum length may affect the rotational moment of inertia for skull rotations and could thus have functional relevance to labyrinth function, empirical evidence shows that full skull length correlates less with labyrinth size than postrostral skull length in birds and crocodiles (Bronzati et al. 2021).”

Reviewer #3 (Remarks to the Author):

This is a great project, everyone! By and large this manuscript is in really good shape. Most of the issues that I see are in the background info where more clarity is required:

- 1) Go through all the references carefully. I don't know all of them by heart, but I do many of them and find that there are quite a few that are inaccurately applied/attributed.
- 2) Your data are awesome enough on their own you don't need to continually over-emphasize the pervasiveness and legitimacy of some "aquatic turtles have small canals" hypothesis, which really doesn't formally exist.
- 3) You occasionally slip into the same contradictions as Spoor et al. both in terminology and interpretation and this has a few significant impacts

Response : We thank Dr. Sipla or Dr. Georgi (both with the forename Justin) for these encouraging words, and for all the detailed comments. In the following, we address the three main points raised separately:

Regarding point (1), we critically reviewed our literature citation and amended some of them. As the reviewer included many individual citation criticisms in their list of detailed comments, we address them one by one below. We thank the referee for this attention to detail, and found these comments particularly helpful.

Regarding point (2): Okay, agreed. We considerably de-emphasized the 'small labyrinth for aquatic taxa' hypothesis, and also don't sell the finding of large turtle labyrinths as quite as surprising anymore (e.g. avoiding use of the word 'surprising').

Regarding point (3). We believe that the reviewer refers to some points they specifically raise in their detailed comments below, and address them individually below. However, we can already note that we followed many of the suggestions, including a more careful approach to labyrinth 'sensitivity'.

I present for your consideration also, a few additional points that should be addressed either by further analysis (in some cases easy, in some admittedly not), or at least direct address in the ms.

Specific comments:

In 20: "predictable effects on sensitivity". 'Sensitivity' here and throughout is always a problematic choice of words, even if it follows that of earlier works. In my view, many of the canal hypotheses that have been falsified over the years suffered because the idea of combined 'sensitivity' of the canal is not meaningful. By the time this ms gets to the end, it teeters on the same precipice by over emphasizing some perceived magic meaningful connection between canals "size" and "sensitivity" in the comparison between turtles and birds. Canal response/efficacy is a matter of both response time and signal discrimination and changes in some parameters can have opposite effects on those two functions. Is a canal more "sensitive" if it's discriminatory capability goes up but it's response time decreases? I realize you don't have the space in this ms to go into all the biophysics, but the introduction and the interpretation of the results will be improved if you make this idea more clear.

Response : This is a very good point to raise by the reviewer, but a hard one to incorporate into the MS without significantly increasing its scope and length. We sourced the literature for an alternative word choice for ‘sensitivity’, and now use ‘response profile’ instead (Georgi et al. 2013), whilst clarifying that the frequently used ‘sensitivity’ can be decomposed into distinct effect that relate to response time and signal discrimination, as described in the equations of Rabbitt et al. (2004). We hope this is a suitable solution. The respective introduction sentence now reads: “The lengths and pathways of the semicircular ducts have predictable effects on the response profile, which is often described as ‘sensitivity’, but can be decomposed into distinct effects such as response time to rotational accelerations, and signal discrimination (Wilson & Melville Jones, 1979; Oman et al., 1987; Spoor & Zonneveld, 1998; Rabbitt et al. 2004; Georgi & Sipla, 2008; David et al. 2010; Georgi et al. 2013).”.

In later instances in the MS, we now specify ‘sensitivity’ further – for instance, when we talk about semicircular duct length (which affects signal discrimination but not response speed), we use ‘sensitivity to signal discrimination’. This is slightly more clunky, but more precise.

In 18: "reflexes that stabilize the visual field" Careful about overstating the role of the canals in visual stabilization. Vestibular signals are important primarily because of their low latency, not because they are the perfect solution to the problem of visual stability. For truly visual species especially, opto-kinetic signals are the real workhorses here, but they are slow enough that the fast vestibular reflexes get things moving in approximately the correct direction first. There's no question VOR and VCR are important, but if they are not the ultimate arbiters of a stable field of view then perhaps moderate to low correlations between vestibular morphology and locomotion are not too surprising.

Response : Thanks for this comment. This is obviously true, and the vestibularly induced reflexes are only one (important) part of eye stabilization. We slightly changed the sentence and now state that VOR and VCR help in stabilizing the visual field, whereas our previous expression suggested they were the only reflexes important here. We also mentioned ‘other functions’ for balance. The sentence now reads as follows: “These provide sensory input on head motion that, among other functions, enable the vestibulo-ocular (VOR) and vestibulo-collic (VCR) reflexes that help stabilize the visual field during locomotion (Steinhausen, 1933; Wever 1978; Wilson & Melville Jones, 1979; Spoor & Zonneveld, 1998; Rabbitt et al., 2004; Georgi & Sipla, 2008; David et al. 2010).”.

In 37: I disagree with the choice of Spoor and Zonneveld to exemplify the correlation between canals and locomotion. 1) The only locomotor correlation they claim to demonstrate is human vs all other primates, 2) the actual statistical support for this is dodgy at best, 3) they outright state in their conclusions that this correlation is not strongly supported, 4) they further state in their conclusions that there are many other more important factors in the morphology of human canals than the flimsy locomotion connection.

Response: We agree with these statements, but the confusion comes from us having cited the incorrect study here. We should have cited Spoor et al. 2007, in which the focus is primates+other mammals. In that study, they explicitly state that “semicircular canal radius of curvature is positively correlated with agility of locomotion on primates and other mammal”, demonstrated by “phylogenetically informed statistical analyses”. Given that this statement and the content of the 2007 paper clearly support our sentence in line 37, we simply updated the citation.

In 38: I've got no complaint about presenting Spoor's agility-based data (it will always remain an important historical step in this field), but as with his use of sensitivity, 'agility' is not a very meaningful biomechanical term (as you cover in the discussion). It's clear from the cited works and conversations with him that he is trying for something akin to turning-performance as a metric (hence his insistence on body mass as well, as a determinant of the rotational moment of an entire animal turning).

Response: Thank you for this comment. As you say, we specifically address problems with agility later in the discussion. We have additionally done a minor clarification here early in the discussion: “This result contrasts with the hypothesis that large labyrinth sizes in tetrapods are related to high agility, or turning performance, in birds and some mammals “. We have de-emphasized the agility expectation for turtles in the MS, so that hopefully at least partially addresses the reviewer’s concern.

In 39: Part of your investigation later on is relative canal size, so I strongly recommend against presenting Kemp and Kirk (2014) without some further commentary on the weakness of their conclusions relative to size. I know Roger has used this in his 2017 paper, and, yes, Kemp and Kirk state that eye-size is a primary driver of canal size but my concern is that if you look at their "eye-size" variable, it is residuals from an eye/body mass regression. Without any direct evidence that eye-size is independent of the overall size of the head (unlikely - so there's also the autocorrelation issue here), what they are interpreting as eye-size to support their visual argument is, based on their data, not actually distinguishable from a simple head-size argument which may or may not favor a biomechanical interpretation that directly links canal size to head motion without reference to the function of the visual system.

Response: This is an interesting perspective, and we thank the reviewer for their consideration. It was useful to think about this. This is our view on the statistical analyses presented by Kemp & Kirk (2014): Their best-supported explanation of labyrinth size, based on AICc, includes body mass, eye size, and visual acuity (“Measurement of visual acuity for 33 species were taken from published sources listed in Table 1”, p. 782). Let’s agree that the independent effect of eye size in this model is just indexing relative head size after accounting for body mass, and can therefore be ignored. After that, we still have an independent effect of visual acuity (after accounting for body mass and relative head size), which explains 11% of the measured vestibular variation (fourth model in Table 3). Thus, we don’t think their conclusions are weak with relation to size. We do agree that the effects of eye size and visual acuity have not been tested for other groups. However, the same is true for the ‘agility’ hypothesis, which also is largely grounded on mammalian work. We checked that our text represents these interpretations correctly, and

refers to ‘visual acuity’ rather than ‘eye size’. We also changed ‘thought to be’ to ‘may be’ to reflect the potential for uncertainties. The sentence now reads as follows: “Proportionally larger labyrinths in agile mammals (Spoor & Zonneveld, 1998; Spoor et al., 2007) may be functionally related to higher visual acuities required for more precise gaze stabilization (Kemp & Kirk 2014).”.

In 50: The three papers cited other than Benson et al all find statistical support of one form or another for locomotor correlation and Hanson et al in some ways directly contradicts Roger's earlier work. Does this really demonstrate "little statistical support"?

Response : We agree that the sentence was poorly constructed, as it suggest the cited studies find no statistical significance of locomotor behaviour on shape. The point we tried to make is that the measured effects are weak at best, and particularly weak compared to the effect of phylogenetic signal (or allometry). Overall, this indicates that locomotor behaviour may be of little importance to structuring labyrinth shape variation. We exchanged our previous sentence by this one: “For example, statistical analyses detect only weak effects (if any) of locomotion on labyrinth shape in birds or extinct reptiles, indicating that locomotion may play a minor influence on labyrinth shape variation.” Even the data in Hansen et al. (2021) support this – R^2 of ecological effects range between 0.1–0.11, with most being <0.05 (their Suppl. Table 4). Thus, even though they claim that ecology is super important in determining labyrinth shape, ecology in their models at best explains 11% of shape variation. It should also be noted that Hansen et al. (2021), with the exception of one model, only use simple regressions of one ecological variable regressed against labyrinth shape, without testing for dependent or independent effects of other ecological variables or variables encoding skull shape in multiple regressions (despite the fact that their 2B-PLS analyses show significant correlations of skull and labyrinth shape). Thus, their effect sizes of ecological variables may very well be even smaller than reported if they are partially redundant with those types of untested effects.

In 64: I see no references for these "studies", but any study that suggests turtles do or should have small canals should be taken with several very large grains of salt as it indicates some very fundamental misunderstanding of turtle canal morphology and basic vestibular principles. If you mean that this is indirectly implied by studies which follow the Spoor model of ponderous locomotion = small canals, then please rethink this, as the different claims of size-to-"agility" relationships in Spoor's model are not even internally consistent (positive general correlation means bigger/less "agile" animals have larger canals...but somehow positive residual on same model equates to more "agile" animal? This has never been adequately explained or defended) let alone particularly broadly applicable.

Response: We cited the reference for ‘these studies’ in the next sentence, which was concerned with details of the general statement of the previous sentence. We have now re-arranged the sentence (shortened and combined them), to make sure what is meant and who is cited for which statement. Specifically, we cite Lautenschlager et al. (2018), who wrote “when compared to more agile reptiles, all turtles possess short canals”; p. 11).

In 117: This is a completely inaccurate citation. Nowhere in that chapter do we claim that aquatic reptiles have overall small canals (absolute or relative); nor would I ever do so (see comment for In 64 above). There are, in fact, no claims of any kind about overall size of canal systems in that chapter as it is entirely based on size independent shape analysis. Actually, the very surprising thing is the small size in the testudines. Given the "free-fall" portion of their step-cycle as the CoM crosses the diagonal line of support (Zug 1971) I would expect larger canals (though the thickness of the canals might be sufficient adaptation here). Perhaps the idea that is being referenced here is merely the statement of eccentricity of the canal circuit in aquatic taxa where the area enclosed by the circuit is smaller *relative to the canal perimeter* than in terrestrial taxa a claim further supported by Goyens analysis.

Response: We apologize for this inaccurate citation, which was based on the notion of smaller aspect ratios in aquatic reptiles – which we agree does not necessarily equate to smaller labyrinth sizes in terms of canal lengths. Nonetheless, several studies have observed small labyrinth sizes for aquatic representatives of various amniote lineages that include aquatic and non-aquatic forms. We modified our sentence to cite these reports more directly by including the taxon for which this was observed (e.g. “eosauropterygians: Neenan et al. 2017”).

In 120-124: This is fascinating. I admit I'm a little surprised by this finding, but only a little as I would expect the length of the neck to be far less relevant than the cross-sectional area of the neck. Neck length does represent some small contribution to the moment of inertia of the head/neck complex, but it's not capturing the capability of neck stabilizing muscles to reduce/counter rotations. I realize that muscle area is probably not a feasible parameter to collect but I would encourage the addition of some other proxy for neck muscle mass - perhaps average of (width * height) across the cervical vertebral series is possible on the specimens where you have a neck length value.

Response : We were also surprised to find no effect regarding our neck measurements. It is possible (and we acknowledge this) that neck length is not the ideal measurement to test for neck movability. It is not feasible to collect addition neck data for this project, as the neck length data was collected from dry skeletal material without associated musculature. However, we think that it performs reasonably in turtles with the apomorphically rigid, shelled body, as long-necked species are largely active predators with a wider range of neck motions, whereas short-necked species often use less elaborate head movements and sometimes even lost the ability of neck retraction (e.g. chelonoids).

In 151-153: See comments in previous section. I don't believe the large size of turtle canals should be characterized as surprising at all, nor should any significant stock be placed in the assumption of small size based on slow locomotion.

Response : I changed the sentence from “This contrasts expectations based on the low overall agility of turtles” to “This contrasts with what would be expected under a general ‘agility’ hypothesis for explaining labyrinth size variation among tetrapods, or even just among reptiles”. We hope this reflects that small labyrinths would be expected if the

'agility' hypothesis were correct. This should be an improvement over the previous wording which implies that the agility hypothesis was widely accepted in the first place, such that deviations from it would be 'surprising'.

In 169: I am curious what happens if you leave out the sea turtles. In the 2008 chapter we also found no significant shape difference within our entire turtle sample but, when we left out the chelonids due to their unusually derived vestibular morphology we uncovered a strong signal within the remaining turtles that converged on the signal within the other amniote groups. This is especially true in your dataset as I strongly doubt that the grade of forelimb webbing truly captures the magnitude of difference between chelonids and other turtles (Pace looked at a flippered trionychid). I suggest that the inclusion of sea turtles here is masking several more important findings. This should at least be discussed but better would be a separate, sea turtle free, analysis.

Response: Thanks a lot for the suggestion to run an analysis excluding sea turtles. As sea turtles are among those turtles with the most 'unusual' morphology, it is indeed interesting to explore any background patterns that may be concealed by including them. I implemented the suggestion in two ways: I re-ran the labyrinth shape `procD.pgls` analyses twice, once excluding all marine turtles (including thalassochelydians, marine bothremydids, stereogyine podocnemidids), and once excluding only chelonoids (i.e. modern sea turtles). Both analyses and results are reported in the supplementary material, but the results do not change much: in both exclusion analyses, the ecological effects (as well as the functional neck variables) have no independently significant effects in multiple regression models that include allometric effects alongside the braincase aspect ratio effect. Thus, even these analyses lead to the same main conclusion that ecological effects (as measured by our variables) do not significantly explain shape variation observed among turtles.

In 171: Again, many of these citations are inaccurate in this context:

- 1) In my 2008 work it explicitly states that "Canal shape is, to a large extent, phylogenetically conservative" and the locomotor adaptation is "secondary variation".
- 2) Malinzak et al. discuss primarily the angles between canals, a feature not even addressed in this ms so I'm not sure it's really fair to critique any assumptions they may or may not have made (and a primacy of locomotor related canal variation is certainly not an explicit assumption in the paper).
- 3) Our 2013 work only ever discusses "adaptive" changes and make no claims of the magnitude of these changes with respect to phylogenetic base-line shape.
- 4) Grohe et al. explicitly test, find, and state several times that phylogenetic signal in their mustelids is stronger than the locomotor signal.
- 5) Eric, to my recollection makes no such claim in his 2016 paper.
- 6) OK....possibly Goyens...

Response: It seems that this review comment includes criticism toward two aspects of line 171: That we say that locomotion has been seen as the primary influence on labyrinth shape by many studies, and that we criticize studies for neglecting phylogenetic

information when they possibly haven't. To disentangle both, we have inserted a full stop to separate both parts of the sentence, which may have been a bit packed before.

Regarding the first point: We agree that our word choice "Locomotor ecology has been assumed to be the primary influence on labyrinth shape" was unlucky, as many studies acknowledge a phylogenetic influence on labyrinth shape that may even be larger. We changed our wording, and the sentence now reads "Locomotor ecology has been assumed to have a notable influence on labyrinth shape". We believe that many of our cited studies do indeed put strong emphasis on the locomotion-labyrinths shape relationship, and thus keep these citations in the first part of the sentence, although we delete the Ekdale 2016 citation, as this is more of a review paper that does not provide independent data that would suggest that ecology has a large effect on labyrinth shape.

Regarding the second part of the sentence, we may have been a bit unclear about what we meant when we wrote that "many analyses concluded this from analytical methods that neglect phylogenetic information". It is true that many studies acknowledge 'phylogenetic signal' of some kind (often k-mult estimates) in labyrinth evolution. However, this is a measure of phylogenetic signal that is agnostic to its potential explanations, and is not independent of the potential that phylogenetically-correlated labyrinth shape variation is driven by phylogenetically-correlated locomotory variation. Similarly, a non-phylogenetic test of correlation between labyrinth shape and locomotor traits may recover a strong correlation, but that correlation can result entirely from the fact that both traits are evolving on the same phylogeny, independent of the existence of any functional linkage between them. Therefore, analyses need to ask whether there is - any- correlation between labyrinth shape and locomotion after accounting for phylogenetic signal. Phylogenetic regression methods specifically ask this question.

What we had intended to communicate, is that most previous studies have based their conclusions of a link between locomotion and labyrinth shape on non-phylogenetic regressions. The Grohe paper is actually a good example – they measure k-mult, which finds phylogenetic signal, but then largely perform non-phylogenetic analyses. For their shape-ecology regressions, they perform both non-phylogenetic ANOVA and pgl's. The regular ANOVA indicates a correlation between locomotion and labyrinth shape, but this effect becomes insignificant when phylogeny is taken into account. This means that the apparent correlation between these traits resulted from the fact that they evolve, independently of each other (i.e. uncorrelated evolution), on the same phylogeny. Yet, the authors state : "However, the phylogenetic ANOVA fail to show any significant ecological signal, meaning that the shape of the bony labyrinth in the mustelid clades 4, 5 and 6 (Fig. 3, Table 7) is explained both by species ecology and their shared evolutionary history." (Grohe et al. 2013, p.11). However, their analyses indicate that there is no effect of species ecology once shared evolutionary history is taken into account. Their analyses instead suggest that the regular ANOVAs should not be considered, while the phylogenetic regressions indicate no influence of ecology at all. Nonetheless we acknowledge that these citations are somewhat 'opaque' – in the sense that these studies often perform phylogenetic analyses, but do not consider them for their interpretations.

We didn't want to get into a lengthy explanation in the manuscript itself. But we hope it is a lot clearer in the revised version: "Furthermore, many of the interpretations of

previous studies have been based on non-phylogenetic analyses (e.g. Grohé et al. 2015; Hansen et al. 2021), often overlooking the low explanatory power or absence of relationships between locomotion and labyrinth shape once phylogeny is taken into account, even within some of the same studies (see phylogenetic regressions in Grohé et al. 2015 and Hansen et al. 2021)”.

In 174: It is inaccurate to claim that Georgi et al. 2013 neglected phylogenetic information. Although we did run analyses on the raw data, all of the significant findings reported were PICs and even the figures are explicitly labeled as showing the regressions and CIs of phylogenetically corrected OLS.

Response : We removed this citation, although the majority of the results reported in that study are based on the Pearson correlations.

In 177-179: Werner Graf was making this argument in response to some of the earliest work back in the 1980's but 40 years later solid data remain elusive. Your numbers tell a compelling story of correlation, but I wonder if your interpretation of the correlation is correct given the size range of the samples...see Goyens for example. I certainly don't think this interpretation has quite the strength it is presented with. In many cases, the flattening of the skull is likely an aquatic adaptation in itself so perhaps this just once again means that the webbing index is not as informative as hoped.

Response: Our analyses show that when these different potential influencers are analysed together, allometry and spatial constraints are returned as much more important than ecology, notwithstanding that ecological variables may be found to have small effects in addition to allometry and spatial constraints (e.g. Bronzati et al. 2021 – but note that study found that ‘aquatic’ variables were non-significant with skull flattening variables when analysed together, whereas skull-flattening variables remains significant). Most of the other literature does not examine the potential effects of skull-flattening on labyrinth shape and so is difficult to compare. In the current MS, we find actually no statistical-significant effect of aquatic adaptation on labyrinth shape in any of our analyses, regardless of what other variables are included. The exception to that is when chelonioids are excluded from analysis (Table S8). There, the result is more similar to Bronzati et al, suggesting that aquatic non-chelonioids -do- on-average have flatter skulls than other turtles. But that skull flattening is much more predictive of labyrinth shape than are aquatic habits (i.e. skull-flattening remains significant after accounting for aquatic habits; but the reverse is not true).

If our measured braincase aspect ratio effect would be nearly the same as an ‘aquatic effect’, we should recover both as statistically significant when analysed on their own, and they should be partially redundant when analysed together. However, this is not what we are measuring. We agree that the braincase aspect ratio and labyrinth shape correlation need further scrutiny, but for now we simply represent our correlations as we find them and advocate for a stronger focus on including spatial constraint variables in future studies, which really is our main point of advocacy. This is important. For example, Hansen et al. (2021) dismissed both spatial constraints and allometric effects as being unimportant, but their allometric analysis has an effect size twice as large as the

largest significant ecological effect in phylogenetic regressions, and although their 2B-PLS analysis recovers a clear and significant correlation of braincase shape and labyrinth shape. The authors based their ecological interpretations on regressions that only include ecological variables without serious consideration of the independent effects of different types of variables, and were thus almost pre-determined to find them to be important.

We added two sentences to this regard to the paragraph, as we feel this is an important discussion that was maybe covered too shortly in our previous MS version. These read as follows: “Hansen et al. (2021) have recently dismissed the effect of allometry and spatial constraints on labyrinth shape for reptiles and birds. However, allometric effects on labyrinth shape (body mass vs. labyrinth shape $R^2 = 0.22$; Hansen et al. 2021) are twice as explanatory as the most important ecological variables in their phylogenetic regressions ($R^2 = 0.11$; Table S4 of Hansen et al. 2021), and their analyses did not investigate the independent effects of ecological effects compared to braincase architecture.”

In 188 and on: I strongly suggest that many of these direct comparisons between turtles and other groups be qualified a little (ok, if it were me, a lot). Yes the relative size comparison is eye-catching but things get too dicey when you begin to compare turtle canal response directly to other animals, especially if you are just using one (less critical than several others) parameter. There are just too many weird aspects in the turtle system:

- 1) the papilla neglecta in turtles seems to encode some semicircular duct flow signals from the common crus (Brichta and Goldberg 1998), that's going to significantly alter neural response.
- 2) Nothing else that I have seen (other than possibly a few ankylosaurs) has the fat-sausage canals of some of these turtles. From a neural response standpoint, the potential for a large duct radius to swamp out nearly all meaningful variation in duct length is enormous.
- 3) Utricular and saccular modifications in some (many?) of these taxa may well have an impact on flow within the duct circuits, especially in light of the extreme duct cross-sections. See comment for In 298-300

As in so many other areas, when turtles fail to fit a larger pattern the default assumption must be that it is turtles that are divergent not that the pattern itself is less applicable.

Response : Regrading the last sentence of the reviewer's comment, this is an interesting way to look at it, and we don't disagree – turtles don't fit the pattern, and that likely results from variation in the soft anatomy of the sensory organ. But other groups also show various differences in soft tissue anatomy of the sensory organ, and the fossil record shows independent increases in labyrinth size in turtles, lepidosaurs and birds. Those aspects are documented in our paper. So what is happening here? Strictly speaking we don't know, but it is likely that there are other 'hidden' aspects of soft tissue variation at play. At what point do we transition from saying 'turtles are weird because of ...', to saying 'variation in ... overwhelms the correlation between osteological size and factors such as 'agility' at large phylogenetic scales'. We believe we should be inclined towards the second way of putting it. We added this sentence to the last paragraph of our discussion (underlined): “Current functional hypotheses related to agility perform poorly in explaining size and shape variation in turtles. Moreover, labyrinth size increases in

stem birds and stem turtles cannot be explained under a single functional hypothesis, indicating that there may be lineage-specific drivers for labyrinth size evolution. Soft tissue variation in the structure and function of the vestibular organ might provide an explanation of this and requires further examination using functional models”.

So we agree that there are many unknowns in turtles and generally amniote labyrinths that can influence neural response. However, and this specifically relates to points 2 & 3, we know that there is no effect of the ‘fat sausage canals’, because the canals are not mirrored in the internal diameter of the semicircular ducts. We know this from segmentation of stained specimens (including *Dermochelys*, cheloniids, testudinids, and other turtles). The huge canals in *Dermochelys* (and other turtles with fat canals) are filled with regularly sized ducts, and there simply is a lot of perilymphatic space around the membranous labyrinth. We have a full separate MS on this, and could not integrate the results of that study into this MS for length reasons (in fact, we cut it out previously). However, we decided to include three sentences to explain this, and show a new supplementary figure that compares the internal duct diameters of two species. The additional sentences read as follows: “Variation in relative internal canal diameter is large among turtles (**Fig. 1**; Evers et al. 2019), and the internal semicircular duct diameter influences labyrinth response time (Rabbitt et al. 2004) independent of duct length or labyrinth size. However, preliminary comparisons of membranous duct diameters to semicircular canal diameters suggest that thick bony canals of sea turtles do not correlate with enlarged duct diameters, but instead accommodate an increased perilymphatic space (**Supplementary Fig. 10**). Therefore, variation in bony canal thickness variation most likely does not have a direct functional relationship to labyrinth response time.”

In 190-193: Your results do indeed contrast with the agility hypotheses, but this is not "unexpected" because those hypotheses are nearly worthless and already counterindicated by several other studies you have already cited up to this point.

Response: Okay, we deleted the part about this being “unexpected”.

In 227: Again, there is NO hypotheses of small size in aquatic animals in Georgi and Sipla 2008 either stated or implied.

Response: We removed the citation.

In 239: Up to this point you have restricted the discussions to amniotes (perhaps because of the emphasis on the claim of small size in aquatic taxa, which is the ms biggest weakness). But if you're going to now postulate a relative large size due to an aquatic habitat you really should bring the fish and their HUGE canals up from the last line of the ms to a fuller discussion right here.

Response: We considerably de-emphasized the claim of small labyrinth size in aquatic taxa, so hopefully we could somehow decrease this weakness of our MS. Regarding the remark to fishes (e.g., teleost fishes, rather than all ‘fishes’ necessarily): It is maybe important to note that we do not suggest ‘all aquatic animals have large labyrinths’, but that aquatic turtles have large labyrinths. Among aquatic groups, labyrinth size seems to

vary considerably (e.g., large in teleost fishes and aquatic turtles, but small in sauropterygians). This only emphasizes the point that we are trying to make: there does not seem to be a generally applicable explanation to labyrinth size patterns in amniotes, and likely vertebrates. However, we agree that the large labyrinth sizes of fishes have so far been neglected from many vertebrate labyrinth size discussions, and so we added a sentence to mention teleost labyrinths as suggested by the reviewer: “It is noteworthy that aquatic turtles are not the only aquatic vertebrates with large labyrinth sizes, as these are also found in teleost fishes (Melville Jones & Spills 1963). However, this association is not generally present among vertebrates because many other aquatic taxa have proportionally small labyrinths (...)”.

In 251-260: This either needs to be fleshed out a little more or dropped completely; as it is presented at the moment it (mostly) contradicts the earlier discussion that the canal morphology is constrained by the brain-case. That is a shape argument and this is a size argument so it is possible that you can find a congruence between the two, but currently it doesn't work (personally I think it's the brain-case argument that is the issue, see comment for In 177).

Response: We do not think that our statements contradict one another. The evolutionary size increase of turtle labyrinths is a temporally restricted, one-time innovation that may cause the otic chamber to increase in size. Other influences on skull shape, such as skull flattening related to feeding modes may then cause the labyrinth shape to be influenced (e.g., lower aspect ratio as available dorsoventral height in braincase decreases). It appears that this wasn't clear enough from our previous text, and so we modified the last two sentences of the paragraph to emphasize the temporal brevity of the labyrinth size increase effect on the braincase: “However, drivers of the initial evolutionary enlargement of otic capsule size, which then necessitated rearrangement of the jaw adductors, have remained unexplained. We propose that increase in labyrinth size at the base of Perichelydia may be the unknown factor causing otic capsule enlargement.”

In 262: The "complexity in labyrinth shape and size evolution" has been recognized since Gray's work at the turn of the 20th century, so I think this needs some clarification.

Response: This is probably true. We modified the sentence, which now says that the story is more complex than ‘implied by some recent comparative studies (e.g., Hansen et al. 2021; but see Georgi & Sipla 2008; Goyens 2019 for more nuanced perspectives)’. We hope that in this way, we can acknowledge that previous studies have recognized labyrinth ecomorphological complexity, but that others have brushed over that and imply relatively straightforward ecological stories.

In 264-266: That's because "size" alone is an awful correlate of actual functional response in a broad taxonomic comparison and really shouldn't ever be used to compare birds and turtles (see comments for In 188). I am 0% surprised that this comparison appears meaningless.

Response : This is true, but it's also true that studies so far have often not looked specifically for lineage-specific drivers of patterns and shape and size evolution for

labyrinths, but instead often have tried to implement what's known for (certain) mammals onto other groups.

In 298-300: I'm concerned about this closed loop of the internal ASC perimeter metric. Many of your specimens, rather than showing extreme morphology of the canals, demonstrate extreme morphology of the vestibule itself driven by utricular and saccular modifications (chelus fimbriatus springs to mind particularly, but it seems you've got some Gopherus and a few other culprits as well). I suspect that this metric is capturing not a little variation from these extremes and not as fully representative of canal morphology as you would like. The problem is bigger in the PSC and LSC, of course, but I don't think the ASC is free of it.

Response : It is true that the inner loop captures some variation not strictly limited to the anterior semicircular canal, but for turtles, this is largely limited to variations in the relative size of the anterior ampulla (see also our new Suppl. Fig. 10). In either case, the 'swelling' of the ampulla region does occur in a range of turtles, but we view it as part of the shape variation of the semicircular canal system. It was also the easiest way to implement some measure of canal thickness. We did leave out the LSC and PSC internal loops because they less reliably depict the inner canal perimeter. As long as our landmark concept captures shape variation of the SCCs (which it arguably does), there is no reason to believe that our methods would not detect semicircular canal–ecology associations, even if the landmark concept includes additional shape features. But regardless of this, I ran the procD.pgls analysis excluding the inner ASC perimeter landmarks. The results are presented in the supplement (and their existence is mentioned in the full text; new supplementary section 2.4 and supplementary Table S9), but they show the same absence of ecological significance as the analyses including the ASC loop.

In 406-416: Yes, using postrostral skull length "avoids effects caused by the elongated rostra" but it also neglects what could possibly be some of the most functionally relevant morphology. How easily the head rotates, or conversely how stable the head is, is fundamentally based on the rotational moment of inertia and that, in turn, is driven by the distribution of mass to the second power, $\sigma (m * r^2)$, giving the rostral skull (the mass most distant from cervical rotations) considerably more functional relevance than the postrostral skull.

Response: This of course makes sense, but our study does not test the functional morphology in beaked animals. Turtles have no elongated rostra, but we were interested in fair statistical comparisons. Given that inclusion of the rostrum in birds (and crocs) would also only increase turtle labyrinth size with regard to birds (and crocs), so that our approach is conservative with regard to our study aims. In addition, full skull length correlates less with labyrinth size even in birds and crocs that show large variation in rostral length (Bronzati et al. 2021), so that our measurement seems justified. We added a respective explanatory sentence to our methods section: "Although rostrum length may affect the rotational moment of inertia for skull rotations and could thus have relevance to labyrinth function, empirical evidence shows that full skull length correlates less with labyrinth size than postrostral skull length in birds and crocodiles (Bronzati et al. 2021).".

To my knowledge ontogenetic scaling of the optic capsule in turtles has not been quantified. However, in every other taxon where the bones of the otic capsule are not fused, there is post-natal growth of the inner ear structures. So much of the results and discussion of this work is focused on the relative size of the inner ear system some this issue should be addressed either in the specimen choice (i.e., are all the specimens at least roughly of mature size) or as an explicit discussion in the results section. I think this is especially important here since some of the larger taxa investigated see growth well over an order of magnitude in length alone.

Response: We actually have that type of data, but we excluded it (together with more analyses about semicircular canal vs. duct thickness) to be published separately. For this paper, it is maybe only important to know that statistical analyses were performed on adult specimens only (we now explicitly state this). But, in case you are interested: Turtle labyrinth show negative allometric growth with regard to skull size, whereas the slopes are consistent between species and very similar to the slope of the inter-specific plot of labyrinth size against skull size (slope around 0.22). For *Dermochelys*, we have specimens ranging from hatchling to large adults, and these are actually included in the PCA morphospace. Adults and juveniles plot very close to one another, indicating there is little shape change over ontogenetic growth. Specimen duplicates for various species are also included in our residual plot, and labyrinth residual sizes are fairly consistent across size ranges, and very similar for individuals of the same size, indicating little intraspecific variation. We do not explicitly detail these results because we are out of space – but plan on a follow up study that digs deeper into these issues.

Except for the few issues mentioned above, the material and methods look good.

Cheers All,
Justin

Reviewers' Comments:

Reviewer #1:

Remarks to the Author:

My personal opinion on this large and comprehensive study is basically the same then the previous submission. My main suggestions haven't apparently add much to the new version, so my general comments are more or less the same.

I understand you don't want to split the article in two or three distinct studies, so my feeling is still that this paper introduces and discusses too many issues, with a large and heterogeneous array of methods. Consequently, many details are lost or unappreciated, in terms of results and discussion. Shape analysis is a powerful method but, nonetheless, anatomy is a matter of details, mostly when considering such a wide taxonomic range.

About PCA, I am rather confused on this point. PCA shows a difference in the distribution, but you say that this is just "apparent", a sort of bias, a false perspective. First, I would recommend being more confident with a PCA output. It is not an inferential method, but a descriptive one and, in this sense, it is therefore more "frank" than other approaches. Second, if you think that the output is just a confounding mirage, why hence do you publish it? It has no sense to publish a result and then saying that this result is not reliable.

About the inferences on the influence on cranial morphology, I am a bit surprised to read that you prefer to avoid speculations on this topic because you lack a proper statistical test. I mean, you publish here a paper investigating a very large set of turtles and other vertebrates, with 63 pages of supplementary information, including results and discussions admittedly related to "allometric and spatial constraints with skull", but feel that inferences on the influence of cranial morphology is too speculative because you have no equation for that?! I am surely the first to recommend caution when presenting evolutionary hypotheses but, in this case, it may look excessive. The ear is tightly integrated in the cranial system, and you recognize that variability can be associated with allometric and spatial constraints of the skull. It looks therefore a bit unusual to skip this topic because there are no maths behind, especially when one takes into account the range of methods and samples handled in this extensive study. Apparently, the same extreme caution you have when dealing with cranial inferences is then lost when you present so many detailed phylogenetic hypotheses on the whole diversity of vertebrates.

So, basically, my opinion on this paper is not changed. This is a very comprehensive and stimulating paper, which is however difficult to fully appreciate because of the too many contents, and because of some mismatch between numbers and biology: from one side we have a PCA that you don't trust, and at the same time hypotheses on cranial influences are avoided because of the lack of a proper statistical test. All this in a paper in which turtles are used to discuss evolution in all tetrapods. A peculiar admixture, indeed.

Reviewer #2:

Remarks to the Author:

This paper is really outstanding in that it uses sophisticated, cutting-edge analyses to examine a huge dataset of turtle inner ear anatomy, resulting in extremely interesting results about the vestibular apparatus of turtles, that are not only interesting in their own right but also bear on other vertebrate systems. The data and analyses alone are important because the sample is from a previously understudied group, and so this provides a much-needed resource for future study. Furthermore, the results are interesting from several perspectives - most importantly, their results upend long-held hypotheses about the relationship between semicircular canal size and so-called 'agility'. The authors have responded to all of my requested revisions and have done a great job with this final version of

the paper. Thank you so much for including citations of my paper as well! I look forward to seeing this in print very soon. I only noticed a few typos/perhaps incomplete sentences that will probably be picked up by other reviewers or editors but I note them below:

Lines 235-236: there may be a repeated word, 'variation'

Lines 300-301: Sentence starting with 'Teleost fishes...' appears to be incomplete.

Best wishes,

Dr. Rachel Racicot, Vanderbilt University

Reviewer #3:

Remarks to the Author:

Thank you for your thorough and well-thought-out responses to my previous comments. All my concerns have been addressed and this revision has been a pleasure to read.

A few small typographic issues I did note:

In 48: Refs not superscripted.

In 290: The Teleost sentence has been abruptly terminated.

Cheers,

Justin Georgi (sorry I wasn't specific last time).

Reviewer #4:

Remarks to the Author:

I really enjoyed this paper. I found the introduction and discussion clear and compelling and appreciated the data visualizations. I found a few of the figures misleading with regards to the model results and have made some suggestions that might help meld the findings with the figures. I also had a question about analysis/data consistency for the cross clade comparisons. But aside from these few suggestions and questions, I think the paper is a nice contribution to the field. I also want to commend the authors for making their CT data available on Morphosource and their scripts accessible.

One suggestion for Figure 1 is to plot the size-corrected shape data and then color by ecology (as already done). I understand why the data are plotted the way they are now and I think the visualizations of the change in canal shape across the PCs are great! But if one of your big findings is a lack of association between canal shape and ecology, the figure is giving a somewhat false impression of that finding, which could lead to false interpretations. If the shape data have such a high association with skull size, perhaps plotting size-corrected shape would more strongly convey your findings.

Alternatively, you could make a figure inset that is similar to Figure 2a but has shape (PC1) instead of canal size and is colored by ecology. This may also convey how ecology and size are confounded in their relationships with canal shape.

Overlaying a phylogeny on the plot (for a phylomorphospace) might similarly be useful to display the correlation between shape and phylogeny but with this much data, that might get too crowded to be useful.

In addition, for Figure 2 it seems like a panel that is identical for panel 2a but is colored by ecology would be really helpful in exploring the finding that relative canal size is related to ecology. However, I really love this figure as-is, so maybe layering on ecology as a trait on the phylogeny tips of panel 2c would also convey this information without disrupting the existing figure too much.

Line 136: You describe $\lambda = 0.890$ as “moderate” phylogenetic signal, but that is a much stronger phylogenetic signal than I see in most studies. I realize that relative phylogenetic signal is hard to interpret but this signal seems strong, not moderate.

Line 290: There is some sort of error here. A sentence trails off with no ending “Teleost fishes have extremely large labyrinths , and heavily..”

Methods: How did you verify that taking different landmark datasets from different papers did not influence the centroid size results used to compare among tetrapod clades? This seems important to verify and explain but I did not see it in the methods.

Dear reviewers,

Thank you for your time and effort to review our MS, which has gotten a lot stronger over the review cycles. Below, please find our point-by-point reply to all reviewer's comments. These are the most important changes implemented at a glance:

- we followed reviewer #4's suggestion to do allometry and braincase aspect-ratio-corrected PCA morphospaces and regression analyses
- we added a new figure, Fig. 2, to show these size-corrected morphospaces, and the effects of skull size and braincase aspect ratio on labyrinth shape
- we updated the GitHub repository and scripts to accommodate code for the new size-corrected analysis
- in response to reviewer #1, we slightly modified Figure 1 to move ecological points in the layering of the figure, to emphasize the overlap of ecological categories. We also inserted dashed ovals to indicate the major overlapping area in morphospace
- in response to reviewer #1, we expanded our discussion on the relationships of braincase shape and labyrinth shape
- following suggestions of reviewer #4, we changed the colour coding for the regression and residual plot in what is now figure 3, so that these indicate ecology (rather than relative labyrinth size, which is information inherent to the plots anyway).
- we correct small typographic and similar issues that the reviewers found
- we changed all formatting requirements to the journal style (including numbering references, including code availability statement etc.)

With best wishes, and on behalf of all authors,
Serjoscha Evers

Point-by-point response:

Reviewer #1 (Remarks to the Author):

My personal opinion on this large and comprehensive study is basically the same then the previous submission. My main suggestions haven't apparently add much to the new version, so my general comments are more or less the same.

I understand you don't want to split the article in two or three distinct studies, so my feeling is still that this paper introduces and discusses too many issues, with a large and heterogeneous array of methods. Consequently, many details are lost or unappreciated, in terms of results and discussion. Shape analysis is a powerful method but, nonetheless, anatomy is a matter of details, mostly when considering such a wide taxonomic range.

Response: We understand this, but it is hard to react to such criticism. In effect, it amounts to "do not write a short paper in this type of journal but write three studies with more length and details, to be published in different journals". Our methodological pipeline is similar to that of other studies of similar length and journal type that focus on macroevolutionary shape changes in large clades. All data, code, and analytical

procedures are provided and explained in our MS and its supplements. The main point of this paper is to assess the broader pattern of anatomical changes and its relationship with function – to achieve statistical power this cannot be done on an anatomical case-by-case basis.

About PCA, I am rather confused on this point. PCA shows a difference in the distribution, but you say that this is just “apparent”, a sort of bias, a false perspective. First, I would recommend being more confident with a PCA output. It is not an inferential method, but a descriptive one and, in this sense, it is therefore more “frank” than other approaches. Second, if you think that the output is just a confounding mirage, why hence do you publish it? It has no sense to publish a result and then saying that this result is not reliable.

Response: PCA is a descriptive method that has merits in indicating the major aspects of shape variation for a given set of shape samples. For example, our PCA clearly shows that particular shapes are located in various areas of the morphospace – for example, labyrinth aspect ratio strongly varies across PC1. It is for these reasons that we show the PCA – i.e., to give an intuitive depiction of the most important variation of labyrinth shape variance. This is important as a first step in summarizing what shape variation is under discussion. It is also expected as a ‘standard’ step in this type of study. Many other studies then go on to make a direct interpretation of the PCA, as though it were a hypothesis. We do not do this, for reasons that we hope now are explained much better in the revised MS (see above). But we still want to include the PCA, and explain why it cannot be interpreted at face-value, not least because we want to discourage future studies from doing so and thereby influence the direction of the field.

We agree that the PCA is a descriptive method that assists in formulating hypotheses, which can be subsequently tested using rigorous statistical means. That is, PCA methods cannot answer the question of whether apparent clustering of ecological groups in the morphospace plot result from a direct functional linkage, or from other effects. In our PCA there is an “apparent” difference in distribution, suggesting an influence of ecology on labyrinth shape. We use follow-up statistical tests of that hypothesis, and surprisingly, we falsify it. We strongly feel that documenting this process in our paper is of significant interest to our readership because of the entrenched idea that labyrinth form and function are tightly linked.

We hope we have explained this more clearly in the revised MS. We also modified Figure 1 in two ways, to visually emphasize the overlap of ecology groups: First, we moved some data points in the layered image to the front, which were previously covered by other points in the densely populated plot. This brings some marine points to the front that were previously covered by other ecologies and vice versa. Second, we included a dashed oval around the central portions of the morphospace in which ecological overlap of two or three ecologies is apparent. Third, we provided a PCA of shape data after correcting for allometry and braincase aspect ratio, showing substantially reduced separation of ecological groups in morphospace.

About the inferences on the influence on cranial morphology, I am a bit surprised to read that you prefer to avoid speculations on this topic because you lack a proper statistical test. I mean, you publish here a paper investigating a very large set of turtles and other vertebrates, with 63

pages of supplementary information, including results and discussions admittedly related to "allometric and spatial constraints with skull", but feel that inferences on the influence of cranial morphology is too speculative because you have no equation for that?! I am surely the first to recommend caution when presenting evolutionary hypotheses but, in this case, it may look excessive. The ear is tightly integrated in the cranial system, and you recognize that variability can be associated with allometric and spatial constraints of the skull. It looks therefore a bit unusual to skip this topic because there are no maths behind, especially when one takes into account the range of methods and samples handled in this extensive study. Apparently, the same extreme caution you have when dealing with cranial inferences is then lost when you present so many detailed phylogenetic hypotheses on the whole diversity of vertebrates.

Response: We are unsure on how to address this comment. In order to test for an influence of labyrinth shape on cranial shape, we would have to measure (i.e., 3D landmark) crania and then perform the opposite regression models of those that are central to this paper. These would be skull shape ~ allometric variables + ecology + labyrinth shape. So, our avoidance of speculating on the effect of labyrinth shape on skull shape is not a matter of lacking "an equation", but of lacking the skull shape data. It would certainly be possible to landmark all skulls and run these regression models in opposite direction. But given the reviewer has already stated that this paper contains too much information, we think that the reviewer would agree that this is best addressed in a separate study.

The original comment by the reviewer regarding this was open-ended, asking us to comment on the specific factors associated with turtle temporal anatomy, and asking a question about whether some of our findings had something to do with the anapsid condition, which we don't think is the case. Now the comment seems to be aimed more about the question if the labyrinth drives cranial shape evolution?

In response to this, we have now modified the section that addresses this claim as published by Hansen et al. 2022 (i.e., "there is no spatial constraint of the braincase on labyrinth shape and instead labyrinth shape drives shape evolution of the braincase"): "Hansen et al.³⁵ instead suggested that ecological signal exerts strong functional selection on labyrinth shape, and that resulting variation in labyrinth shape causes variation in adult braincase morphology due to developmental linkage^{35,49}. This hypothesis is causally the opposite of a 'spatial constraints' hypothesis, in which variation in braincase morphology is under stronger functional selection and causes variation in labyrinth shape³⁴ (also due to developmental linkage⁴⁹). We regard the spatial constraints hypothesis as being more plausible for several reasons: (1) published analyses so far, including ours, report at most a weak correlation of labyrinth shape to ecological traits^{18,21-22,34}, suggesting little functional selection for labyrinth shape; (2) in the few cases examined so far, the effects of braincase shape remain after accounting for ecological effects, but the effects of ecological traits do not remain after accounting for braincase shape³⁴, suggesting primacy of selection on braincase shape over any potential ecological influence on labyrinth shape specifically; and (3) similar or identical functional performance can result from highly different-shaped labyrinths⁴⁷, such that selection on functional performance of the labyrinth should not require very specific shape outcomes of the type that might over-rule selection on braincase morphology."

We hope that this section is a bit more like what reviewer 1 was expecting to see?

So, basically, my opinion on this paper is not changed. This is a very comprehensive and stimulating paper, which is however difficult to fully appreciate because of the too many contents, and because of some mismatch between numbers and biology: from one side we have a PCA that you don't trust, and at the same time hypotheses on cranial influences are avoided because of the lack of a proper statistical test. All this in a paper in which turtles are used to discuss evolution in all tetrapods. A peculiar admixture, indeed.

Response: We acknowledge this opinion, but don't see the mismatches the reviewer raises. In our modified MS version, we now show the shape deformations for the association between labyrinth shape and braincase aspect ratio (Fig. 2), which are related to the effects of braincase geometry; and we also expanded the discussion of effects between braincase shape and labyrinth shape.

Reviewer #2 (Remarks to the Author):

This paper is really outstanding in that it uses sophisticated, cutting-edge analyses to examine a huge dataset of turtle inner ear anatomy, resulting in extremely interesting results about the vestibular apparatus of turtles, that are not only interesting in their own right but also bear on other vertebrate systems. The data and analyses alone are important because the sample is from a previously understudied group, and so this provides a much-needed resource for future study. Furthermore, the results are interesting from several perspectives - most importantly, their results upend long-held hypotheses about the relationship between semicircular canal size and so-called 'agility'. The authors have responded to all of my requested revisions and have done a great job with this final version of the paper. Thank you so much for including citations of my paper as well! I look forward to seeing this in print very soon.

Response: Thank you very much for these kind words.

I only noticed a few typos/perhaps incomplete sentences that will probably be picked up by other reviewers or editors but I note them below:

Lines 235-236: there may be a repeated word, 'variation'

Response: Thanks for catching this, second instance of 'variation' is deleted.

Lines 300-301: Sentence starting with 'Teleost fishes...' appears to be incomplete.

Response: Ups, this is a remnant of a sentence that was moved up in the previous MS correction. We deleted these extra words.

Best wishes,
Dr. Rachel Racicot, Vanderbilt University

Reviewer #3 (Remarks to the Author):

Thank you for your thorough and well-thought-out responses to my previous comments. All my concerns have been addressed and this revision has been a pleasure to read.

Response: Thanks again for this very thoughtful review. We are glad you like our changes.

A few small typographic issues I did note:

In 48: Refs not superscripted.

Response: Thanks for catching this. We corrected the text.

In 290: The Teleost sentence has been abruptly terminated.

Response: Ups, this is a remnant of a sentence that was moved up in the previous MS correction. We deleted these extra words.

Cheers,
Justin Georgi (sorry I wasn't specific last time).

Reviewer #4 (Remarks to the Author):

I really enjoyed this paper. I found the introduction and discussion clear and compelling and appreciated the data visualizations. I found a few of the figures misleading with regards to the model results and have made some suggestions that might help meld the findings with the figures. I also had a question about analysis/data consistency for the cross clade comparisons. But aside from these few suggestions and questions, I think the paper is a nice contribution to the field. I also want to commend the authors for making their CT data available on Morphosource and their scripts accessible.

Response: We thank reviewer #4 for the very positive comments. The details of their suggestions are outlined below, and we respond to them individually below.

One suggestion for Figure 1 is to plot the size-corrected shape data and then color by ecology (as already done). I understand why the data are plotted the way they are now and I think the visualizations of the change in canal shape across the PCs are great! But if one of your big findings is a lack of association between canal shape and ecology, the figure is giving a somewhat false impression of that finding, which could lead to false interpretations. If the shape data have such a high association with skull size, perhaps plotting size-corrected shape would more strongly convey your findings. Alternatively, you could make a figure inset that is similar to Figure 2a but has shape (PC1) instead of canal size and is colored by ecology. This may also convey how ecology and size are confounded in their relationships with canal shape.

Response: These are very helpful comments, and we thank the reviewer for them. We agree that size-corrected shape data is appropriate given the allometric effect and the effect of the braincase aspect ratio. However, rather than substituting the original (non-

size corrected) PCA data, we opted to include a new figure, Figure 2. Ultimately, we think that both size-corrected and un-corrected data should be shown. The new PCA uses shape data after correction for skull size and braincase aspect ratio.

We also really liked the idea of plotting shape against skull size or size aspect ratio, and so we opted to also include this, too, into our new Figure 2. This figure now consists of six panels. The top row includes the size-corrected PCA morphospaces, analogue to Figure 1. The middle row shows bi-plots of shape regression scores against the primary influences on labyrinth shape, i.e. skull size and braincase aspect ratio. The bottom row shows landmark deformation plots to indicate the shape change that is associated with small vs. large skull sizes and low vs. high braincase aspect ratios.

This additional component of corrected shape data warranted the addition of a new methods subheading that explains the procedure of arriving at the plots in Figure 2. We also add brief statements to the result and discussion sections. The corrected morphospaces have a similar data point distribution than the original ones, but with more substantial overlaps among ecological groups, especially on PC1 and PC3. We added some text to explain this. The script for the size-corrected analyses was added to our code repository.

Overlaying a phylogeny on the plot (for a phylomorphospace) might similarly be useful to display the correlation between shape and phylogeny but with this much data, that might get too crowded to be useful.

Response: Given the large sample size and number of points in the PCAs, we think this may indeed overcrowd the figure. Thus, we have not plotted a phylomorphospace.

In addition, for Figure 2 it seems like a panel that is identical for panel 2a but is colored by ecology would be really helpful in exploring the finding that relative canal size is related to ecology. However, I really love this figure as-is, so maybe layering on ecology as a trait on the phylogeny tips of panel 2c would also convey this information without disrupting the existing figure too much.

Response: This is true. We experimented with coding ecology information into what has now become Figure 3, but it was difficult to combine the cold–warm colour scheme previously consistently used for the figure with the ecological colour scheme.

Line 136: You describe $\lambda = 0.890$ as “moderate” phylogenetic signal, but that is a much stronger phylogenetic signal than I see in most studies. I realize that relative phylogenetic signal is hard to interpret but this signal seems strong, not moderate.

Response: We modified ‘moderate’ to ‘strong’, as you are right.

Line 290: There is some sort of error here. A sentence trails off with no ending “Teleost fishes have extremely large labyrinths , and heavily..”

Response: Yes, the other reviewers also caught this error, which we corrected.

Methods: How did you verify that taking different landmark datasets from different papers did not influence the centroid size results used to compare among tetrapod clades? This seems important to verify and explain but I did not see it in the methods.

Response: Although some of the landmark dataset originate from different studies, we were still the ones who produced them (for those other studies). Nevertheless, centroid size data was checked and verified to avoid any type of unit errors that could affect results. To be explicit about this in our methods, we added the following sentence: “All landmark datasets were combined and inspected, and measurement units were adjusted to mm for all specimens across the datasets.”

Reviewers' Comments:

Reviewer #4:

Remarks to the Author:

I have no further comments.